# T5SCORE: Discriminative Fine-tuning of Generative Evaluation Metrics

**Yiwei Qin**♣    **Weizhe Yuan**♠    **Graham Neubig**♣♡    **Pengfei Liu**♣♡∗

♣Carnegie Mellon University, ♠New York University, ♡Inspired Cognition
{yiweiq,gneubig,pliu3}@cs.cmu.edu   wy885@nyu.edu

## Abstract

Modern embedding-based metrics for evaluation of generated text generally fall into one of two paradigms: *discriminative* metrics that are trained to directly predict which outputs are of higher quality according to supervised human annotations, and *generative* metrics that are trained to evaluate text based on the probabilities of a generative model. Both have their advantages; discriminative metrics are able to directly optimize for the problem of distinguishing between good and bad outputs, while generative metrics can be trained using abundant raw text. In this paper, we present a framework that combines the best of both worlds, using both supervised and unsupervised signals from whatever data we have available. We operationalize this idea by training T5SCORE, a metric that uses these training signals with mT5 as backbone.[1] We perform an extensive empirical comparison with other existing metrics on 5 datasets, 19 languages and 280 systems, demonstrating the utility of our method.[2] Experimental results show that: T5SCORE achieves the best performance on all datasets against existing top-scoring metrics at the segment level. [3]

## 1   Introduction

Automatically evaluating the quality of generated text plays an essential role in the development of text generation systems (Lin and Hovy, 2003; Peyrard, 2019; Mathur et al., 2020a). A key element of this evaluation is the design of an automated metric that can *recognize high-quality texts*. The current most-popular approach to create such high-quality metrics is the *discriminative* paradigm.

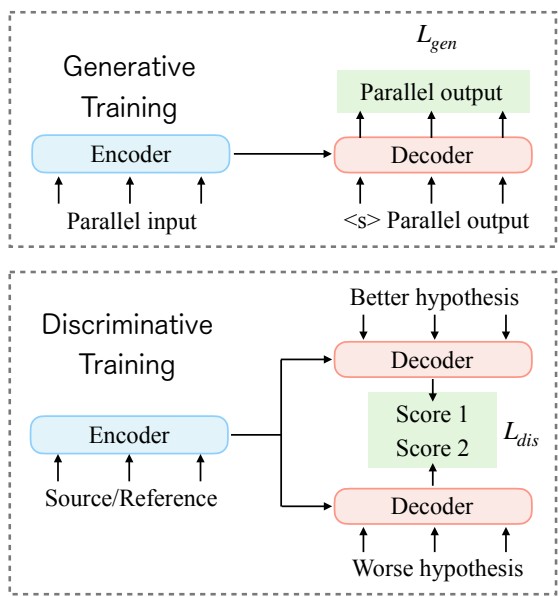

Figure 1: Our framework supports generative and discriminative training. The former uses parallel data and maximizes the probability of output data conditioned on the input data. The latter ranks two hypotheses by their manual scores and maximizes the probability of the better hypothesis while minimizing the probability of the worse hypothesis. $L_{gen}$ and $L_{dis}$ denote generative loss and discriminative loss respectively.

These models are generally trained by taking a sentence embedding model and fine-tuning it using human judgments of generated text quality as a learning signal, allowing metrics to directly predict the quality score of a text. Popular examples include COMET (Rei et al., 2020) and BLEURT (Sellam et al., 2020). However, the effectiveness of this method comes at the cost of expensive manual annotation of human judgements, and thus these models are less broadly applicable than more common lexical metrics such as BLEU (Papineni et al., 2002), ROUGE (Lin, 2004).

More recently, there has been promising work on *generative* metrics. These metrics recognize high-quality texts by formulating evaluation as a

---

∗Corresponding author

[1]We use mT5 because it supports more languages compared to other options (e.g., mBART) and provides different scales of models (e.g., 3B, 11B).

[2]Appendix A.9.1 shows the dataset details.

[3]We release our code and models at https://anonymous.4open.science/r/T5Score-F21D.

generation task and using the generative likelihood as an indication of quality, making it possible to train the models without explicit human annotation. Examples of these metrics include BARTScore (Yuan et al., 2021) and PRISM (Thompson and Post, 2020a). However, because such generative models do not utilize human judgements at training time, these models are inherently at a disadvantage compared to metrics that can utilize supervision.

In this work, we argue that it is crucial to utilize *all* possible supervision symbols that could indicate the quality of the text. To this end, we propose a framework for learning evaluation metrics based on the assumption that *generative and discriminative objectives can work in concert to train a better evaluator*, as shown in Fig. 1.

We achieve this idea by (1) starting with the pre-trained model mT5 (Xue et al., 2021), (2) training mT5 in a generative fashion by maximizing the probability of existing parallel data, then (3) fine-tuning mT5 discriminatively by minimizing a contrastive loss function to teach it to ensure that the generative probability of high-quality texts is higher than that of low-quality texts. At evaluation time, the probability of generating a text is used as the quality score, because the model has learned to assign high probability to superior texts. Our framework has the flexibility to choose from a supervised training strategy and an unsupervised training strategy depending on if human judgements are available for a given language or task, while keeping the evaluation process the same.

We evaluate the proposed metric (T5SCORE) on 5 datasets covering machine translation (MT) and summarization tasks across 19 languages. Regarding reference-based experiments, at the segment level, T5SCORE trained generatively achieves the best performance on one dataset without human annotated training examples; T5SCORE trained discriminatively achieves the best performance on 4 datasets with human annotated training examples against top-scoring counterparts. At the system level, T5SCORE trained discriminatively achieves the best performance in 5 of 6 test settings (2 correlation methods $\times$ 3 datasets). Empirical results also show the effectiveness of generative training, especially for tasks without human judgments. Regarding source-based experiments, we find it better at evaluating top-scoring systems compared to reference-based evaluation, showing the importance of developing source-based evaluation as ma-

chines generate higher-quality texts.

## 2 Task Formulation

Text generation evaluation aims to design a function auto_eval($\cdot$) that takes in a source text $\mathbf{x}$, some reference outputs $\mathbf{y}$ and a system output $\hat{\mathbf{y}}$ and predicts a scalar value that indicates the quality of the system output. The validity of the designed function depends on the degree of correlation between auto_eval($\cdot$) and human judgements (which can be denoted as manual_eval($\cdot$)). The better the correlation between the two, the more effective we consider our designed function to be.

Specifically, in this work, we call an evaluation function (1) *source-based* if it takes only $\mathbf{x}$ and $\hat{\mathbf{y}}$ and predicts using auto_eval($\mathbf{x}, \hat{\mathbf{y}}$) (2) and call an evaluation function *reference-based* if it takes only $\mathbf{y}$ and $\hat{\mathbf{y}}$ and predicts using auto_eval($\mathbf{y}, \hat{\mathbf{y}}$), or it takes $\mathbf{x}, \mathbf{y}$ and $\hat{\mathbf{y}}$ and predicts using auto_eval($\mathbf{x}, \mathbf{y}, \hat{\mathbf{y}}$).

## 3 Metric Design

In this section, we describe T5SCORE and explain how to train the metric in both a generative and discriminative fashion.

### 3.1 Evaluation as Generation

Following Yuan et al. (2021), we formulate text generation evaluation as a text generation problem.

Specifically, the quality of a generated text is measured by calculating the per-token conditional probability of one text $\mathbf{a}$ given another text $\mathbf{b}$, which we also abbreviate as "$\mathbf{b} \rightarrow \mathbf{a}$":

$$\text{T5SCORE} = \frac{1}{|\mathbf{a}|} \log p(\mathbf{a}|\mathbf{b}; \theta) \qquad (1)$$

$\theta$ are the parameters of the sequence-to-sequence model used to calculate these probabilities. Depending on which strings we use for $\mathbf{a}$ and $\mathbf{b}$ we can evaluate the text from different perspectives. We adopt the definition of *Precision*, *Recall* and *F* score based on different generation directions (Yuan et al., 2021):

***Precision*: ($\mathbf{x}$ or $\mathbf{y} \rightarrow \hat{\mathbf{y}}$):** Calculate probability from reference (or source) text to generated hypothesis $p(\hat{\mathbf{y}}|\mathbf{y}; \theta)$ (or $p(\hat{\mathbf{y}}|\mathbf{x}; \theta)$).

***Recall*** ($\hat{\mathbf{y}} \rightarrow \mathbf{x}$ or $\mathbf{y}$):** Calculate probability from generated hypothesis to reference (or source) text $p(\mathbf{y}|\hat{\mathbf{y}}; \theta)$ (or $p(\mathbf{x}|\hat{\mathbf{y}}; \theta)$).

***F* score ($\mathbf{x}$ or $\mathbf{y} \leftrightarrow \hat{\mathbf{y}}$):** The arithmetic average of Precision and Recall to consider both directions.

According to preliminary experiments, *F* score correlated better with human evaluation scores on the DA20 dataset (§4.1) than *Precision* and *Recall*, so we adopt *F* score for default. In order to support multilingual evaluation, we choose mT5 (Xue et al., 2021) as our pre-trained model.

## 3.2 Generative Training for T5SCORE

Generative training aims to teach the model to generate target text from the input text with a standard negative log likelihood loss:

$$\mathcal{L}_{\text{gen}} = -\frac{1}{m}\sum_{t=1}^{m}\log p(\mathbf{y}_t|\mathbf{y}_{<t},\mathbf{x};\theta). \quad (2)$$

We use the MT dataset ParaCotta (Aji et al., 2022) and paraphrasing dataset MT-prism (Thompson and Post, 2020b) as parallel corpora[4] to train our models generatively.

## 3.3 Discriminative Training for T5SCORE

We also design discriminative training methods where human judgments for generation quality are available. Suppose we have an annotated training dataset $\mathcal{D} = \{\mathbf{x}_i, \mathbf{y}_i, \hat{\mathbf{y}}_i, m_i | i = 1, ..., N\}$, where $\mathbf{x}_i$, $\mathbf{y}_i$, $\hat{\mathbf{y}}_i$, and $m_i$ denote the $i$-th example of the source text, the reference text, the hypothesis text, and the manual score, respectively ($\hat{\mathbf{y}}_i$ and $m_i$ can be multiple hypotheses with their corresponding quality scores). We first generate a relative rank dataset $\mathcal{D}_{\text{RR}} = \{\mathbf{x}_i, \mathbf{y}_i, \hat{\mathbf{y}}_i^+, \hat{\mathbf{y}}_i^-, m_i^+, m_i^- | i = 1, ..., N\}$ by finding a pair of hypotheses $\hat{\mathbf{y}}_i^+$ with higher manual score $m_i^+$ and $\hat{\mathbf{y}}_i^-$ with lower manual score $m_i^-$ for the same source text $\mathbf{x}_i$ and reference text $\mathbf{y}_i$. Then, to encourage the model to assign higher probabilities to the better hypothesis $\hat{\mathbf{y}}^+$, we adopt a contrastive loss function, following Liu et al. (2022); Hopkins and May (2011):

$$\mathcal{L}_{\text{dis}} = \max(0, f(\hat{\mathbf{y}}^-) - f(\hat{\mathbf{y}}^+) + \alpha(m^+ - m^-)) \quad (3)$$

where $\alpha$ is the weight of the margin term. $f$ is defined as $f(\hat{\mathbf{y}}) = \frac{1}{m}\sum_{t=1}^{m}\log p(\hat{\mathbf{y}}_t|\hat{\mathbf{y}}_{<t}, \mathbf{y}, \theta)$ for reference-based methods, and $f(\hat{\mathbf{y}}) = \frac{1}{m}\sum_{t=1}^{m}\log p(\hat{\mathbf{y}}_t|\hat{\mathbf{y}}_{<t}, \mathbf{x}, \theta)$ for source-based methods, where $m$ is the number of tokens in $\hat{\mathbf{y}}$.

Because we adopt F score for evaluation by default, our training process also considers two generation directions: from $\mathbf{x}$ or $\mathbf{y}$ to $\hat{\mathbf{y}}$ and from $\hat{\mathbf{y}}$ to $\mathbf{x}$ or $\mathbf{y}$. We augment the training samples by repeating

---

[4]Appendix A.9.2 shows the corpus details.

the corpus $\mathcal{D}_{\text{RR}}$ and changing $\mathbf{x}$ or $\mathbf{y}$ which is originally the model's input to the output and changing $\hat{\mathbf{y}}$ which is originally the model's output to the input. Thus, half of the time we calculate $f(\hat{\mathbf{y}}) = \frac{1}{m}\sum_{t=1}^{m}\log p(\mathbf{y}_t|\mathbf{y}_{<t}, \hat{\mathbf{y}}, \theta)$ for reference based methods, and $f(\hat{\mathbf{y}}) = \frac{1}{m}\sum_{t=1}^{m}\log p(\mathbf{x}_t|\mathbf{x}_{<t}, \hat{\mathbf{y}}, \theta)$ for source based methods.

# 4 Experimental Setup

## 4.1 Evaluation Datasets

We evaluate on 5 datasets: the Direct Assessment (DA) corpus from WMT20 metrics shared task (**DA20**; Mathur et al. (2020b)); datasets obtained by re-annotating the outputs from WMT20 and WMT21 shared task according to the Multi-dimensional Quality Metrics (MQM) framework (**MQM20** & **MQM21**; Lommel et al. (2014)); the dataset of WMT20 shared task on Quality Estimation (**QE20**; Specia et al. (2021)); and a multilingual summarization dataset (**MultiSumm**; Koto et al. (2021)). Details in Appendix A.9.1.

## 4.2 Correlation Measurements

We consider both system-level and segment-level correlations with human judgments when evaluating automated metrics.

**System-level** evaluation calculates average human scores for each generation system to produce a scalar rating for the system performance. We employ the Pearson correlation (sys-p) and Kendall's Tau correlation (sys-k) as the evaluation measure for system-level metrics.

**Segment-level** correlation measures the correlation over segment-level assessments. We keep the same setup as in Mathur et al. (2020b) converting Direct Assessment (DA) to DA relative rank (DARR) and adopting a Kendall's Tau-like (seg-k) formulation as the evaluation measure. We adopt the bootstrapping method (p-value < 0.05) (Koehn, 2004; Graham et al., 2014) for pair-wise significance tests.

## 4.3 Baseline Metrics

We consider the following baseline metrics for comparison: **BLEU** (Papineni et al., 2002) which is the precision of n-grams of the MT output compared to the reference; **ROUGE** (Lin, 2004) which measures the lexical overlap between the system and reference; **COMET** (Rei et al., 2020) which is a discriminative metric that uses XLM-RoBERTa to encode source, hypothesis and reference and can be optimised towards different ob-

jectives; **BERTScore** (Zhang et al., 2019) which computes the cosine similarity between the reference and hypothesis tokens' embeddings based on BERT (Devlin et al., 2018); **BLEURT** (Sellam et al., 2020) which is a BERT-based regression model trained on synthetic examples and ratings from WMT; **PRISM** (Thompson and Post, 2020a) which is a generative metric that scores MT system outputs conditioned on their respective human references; **BARTScore** (Yuan et al., 2021) which is a generative metric that uses BART (Lewis et al., 2019) to evaluate the generated text. [5]

## 5 Reference-based Evaluation

We consider two tasks in reference-based evaluation: machine translation (DA20, MQM20 and MQM21) and summarization (MultiSumm).

### 5.1 Training Details

We consider four different sizes of base models: mT5-B (580M parameters), mT5-L (1.2B parameters), mT5-XL (3.7B parameters), and mT5-XXL (11B parameters). Both generative and discriminative training are considered, with the former based on ParaCotta corpora and the latter based on WMT DA corpora from 2017 to 2019. Our model implementation is based on Huggingface transformers (Wolf et al., 2020). More details of the hyperparameters, training time, computing resources can be found at Appendix A.2.

**DA20** Evaluation of the training models is carried out on the DA20 dataset.

**MQM** We consider various discriminative training data, resulting in the following models:[6]

(a) T5SCORE-*[20] is trained on WMT DA corpus from 2017 to 2019.
(b) T5SCORE-*[21] is trained on WMT DA corpus from 2017 to 2020.
(c) T5SCORE-*[21]$_{mqm}$ is (b) further trained for 1 additional epoch on MQM20.

**MultiSumm** Due to the small size of the MultiSumm dataset (135 examples per language pair), we do not undertake the additional training of a model specific to summarization. Instead, we use models trained on ParaCotta and WMT directly.

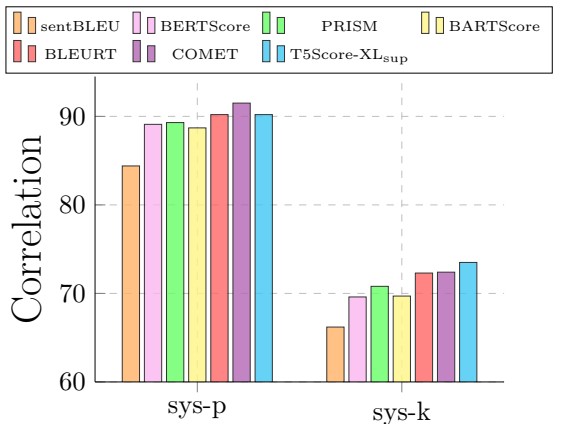

Figure 2: System-level Kendall's Tau and Pearson correlations for the WMT DA20 corpus. Detailed results can be found at Appendix A.4.

### 5.2 Results

For DA20, Tab.1/Tab.7 shows segment level Kendall's Tau correlation results of diverse metrics for 10/8 language pairs with English as target/source;[7] Fig.2 shows system level results on average. For MQM20 and MQM21, Tab.2 shows both segment level and system level results. For Multi-Summ, Fig. 3 illustrates the segment Kendall's Tau correlation.[8][9]

In all tables, ‡denotes correlations not significantly outperformed by any other metric for the given language pair, while †denotes correlations not significantly outperformed by any other unsupervised metric. The highest correlation for each language pair by unsupervised methods is underlined, and the highest correlation overall is bold.

From the above tables and figures, we observe:

1) *At segment level, our method achieves the best performance on average.* Supervised T5SCORE-

---

[5] The specific models used for each baseline Metric can be found at Appendix A.10.

[6] More training results could be found at Appendix A.1.

[7] We find that one MT system NiuTrans.1511 of language pair en-zh generates unnecessary spaces between Chinese characters. Most evaluation metrics cannot handle the spaces well, causing an obvious outlier MT system and influencing correlation greatly, especially Pearson's correlation which is notably sensitive to outliers. To remove the influence, we delete the spaces between Chinese characters of MT system's outputs at evaluation time for every evaluation metric.

[8] For simplicity, we report $F$ score of our model and other baselines like BERTScore, BARTScore which also differentiate *Precision*, *Recall* and $F$ score.

[9] Because segment scores are reliable only when averaged over sufficient number of judgments (Mathur et al., 2020b), we choose Kendall's Tau-like segment score as our correlation metric, different from the original paper (Koto et al., 2021), which uses segment Pearson correlation. We also report Pearson correlation results in Appendix A.8 for reference.

Table 1: Segment-level Kendall's Tau correlations for the WMT DA20 corpus. Avg-en denotes the average correlation achieved by a metric across all x-en language pairs. Avg-x denotes the average correlation across all en-x language pairs, and Avg denotes the average correlation across all language pairs. This table shows the results of all x-en language pairs and the results of en-x language pairs can be found at Appendix A.3.

| | cs-en | de-en | iu-en | ja-en | km-en | pl-en | ps-en | ru-en | ta-en | zh-en | Avg-en | Avg-x | Avg |
|---|---|---|---|---|---|---|---|---|---|---|---|---|---|
| UNSUPERVISED METHODS | | | | | | | | | | | | | |
| sentBLEU | 6.8 | 41.1 | 18.1 | 18.8 | 22.6 | -2.5 | 9.6 | -0.5 | 16.3 | 9.3 | 13.9 | 30.0 | 21.1 |
| BERTScore | 11.7 | 45.2 | 21.6 | 24.3 | 27.9 | 4.7 | 15.9 | 6.0 | 21.9 | 13.4 | 19.3 | 39.3 | 28.2 |
| PRISM | 13.5‡ | 46.5 | 25.5 | 26.3 | 30.4 | 6.6 | 16.5 | 10.0 | 23.0 | 14.5 | 21.3 | 41.9 | 30.5 |
| BARTScore | 12.4 | 48.5† | 23.5 | 26.6 | **31.8‡** | 9.1† | 16.0 | 12.8† | 23.8 | 16.3† | 22.1 | 43.4 | 31.6 |
| T5SCORE-B$_{un}$ | 12.9 | 48.4 | 24.3 | 26.0 | 30.4 | 8.3 | **19.4‡** | 11.9 | 24.0† | 15.9 | 22.2 | 42.8 | 31.3 |
| T5SCORE-L$_{un}$ | 13.0† | 48.7† | 26.6† | 27.9† | 30.9 | 8.5 | 17.7 | 12.8 | 24.0† | 16.5† | 22.7 | 45.3 | 32.7 |
| T5SCORE-XL$_{un}$ | 13.0 | 48.8† | 26.1† | 27.7† | 29.8 | 9.2† | 17.7 | 13.2† | 23.8† | 15.9 | 22.5 | 46.7 | 33.3 |
| SUPERVISED METHODS | | | | | | | | | | | | | |
| BLEURT | 13.6‡ | 47.6 | 27.1 | 28.1 | 31.2‡ | 4.7 | 18.4‡ | 10.3 | 25.3 | 14.7 | 22.1 | 50.0 | 34.5 |
| COMET | 12.9 | 48.5 | 28.1 | 27.4 | 29.8 | **9.9‡** | 15.8 | **15.6‡** | 24.2 | **17.1‡** | 22.9 | 51.0 | 35.4 |
| T5SCORE-B$_{sup}$ | 13.9‡ | 48.5 | **29.2‡** | 28.1 | 30.3 | 9.6‡ | 17.4 | 13.1 | 23.9 | 15.5 | 22.9 | 47.2 | 33.7 |
| T5SCORE-L$_{sup}$ | **14.0‡** | 49.3‡ | 28.5‡ | 28.9‡ | 30.1 | 8.3 | 17.6 | 15.3‡ | **25.9‡** | 16.3 | 23.4 | 50.4 | 35.4 |
| T5SCORE-XL$_{sup}$ | 12.8 | **49.6‡** | **29.2‡** | **29.1‡** | 31.5‡ | 9.3 | 18.0 | 15.2‡ | 25.4 | 15.8 | **23.6** | **51.4** | **36.0** |

XL surpasses all baselines for DA20, MQM20 and MQM21; unsupervised T5SCORE-L surpasses all baselines for MultiSumm.

2) *At segment level, as language model size increases, the metric performance tends to saturate.* In Tab.7, from T5SCORE-B to T5SCORE-L and from T5SCORE-L to T5SCORE-XL, the performance of our unsupervised metric improves by 1.4 and 0.6 on average, respectively; while the performance of our supervised metric improves by 1.7 and 0.6, respectively.[10] However, this is not so clear at system level. A possible reason is that at the system level, there are usually less than 20 systems to be evaluated, much fewer than the number of examples at the segment level, so tiny differences in one MT system can have a large impact on the final results.

3) *At system level, our method is better at Kendall's Tau correlation compared to Pearson correlation.* In Fig.2, our method achieves the highest Kendall's Tau correlation compared to other baselines while performs slightly worse than COMET in terms of Pearson correlation. This can be attributed to our training process, which adopts a contrastive loss function, making our models better at predicting the relative rank of examples or systems instead of the absolute score.

Figure 3: Segment-level Kendall's Tau correlation on MultiSumm corpus. Details in Appendix A.8.

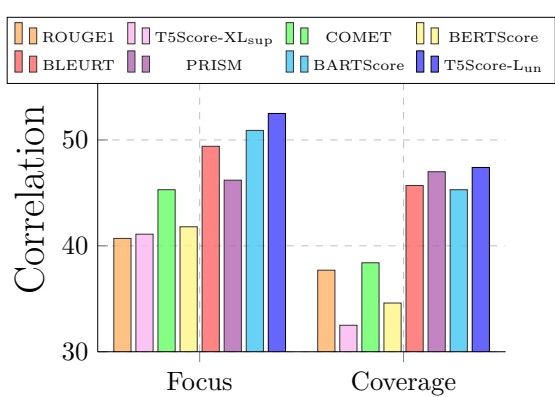

4) *For datasets without human annotated training examples, our unsupervised method achieves the best performance.* In Fig. 3, our supervised methods perform worse than unsupervised methods, and other supervised methods do not work well either. The reason could be that these methods are trained on MT data, and their direct use for the summarization task may impair their performance. These results also indicate that our method has the advantage that the unsupervised version still works well in the absence of human annotated data.

## 6 Source-Based Evaluation

We also support source-based discriminatively trained methods. In this section, we show the effectiveness of the source-based method. We consider the task of machine translation and inspect the re-

---

[10]From T5SCORE-XL to T5SCORE-XXL, the performance of our unsupervised metrics improves by 0.3. Detailed results can be found at Appendix A.5. Due to the limited computational resources and the limited performance improvement of T5SCORE-XXL, we don't train supervised T5SCORE-XXL.

Table 2: Segment Kendall's Tau, system Pearson and system Kendall's Tau of different metrics on MQM20 and MQM21 dataset. Avg. denotes the average correlation achieved by a metric across two language pairs and two years. Method COMET uses model wmt20-comet-da and wmt21-comet-mqm for MQM20 and MQM21 respectively. Method T5SCORE-XL$_{sup}$ uses model T5SCORE-*[20] and T5SCORE-*$_{mqm}^{21}$ for MQM20 and MQM21 respectively.

| | MQM-2020 | | | | | | MQM-2021 | | | | | | avg | | |
| | en-de | | | zh-en | | | en-de | | | zh-en | | | | | |
| | sys-p | sys-k | seg-k | sys-p | sys-k | seg-k | sys-p | sys-k | seg-k | sys-p | sys-k | seg-k | sys-p | sys-k | seg-k |
| UNSUPERVISED METHODS | | | | | | | | | | | | | | | |
| sentBLEU | 82.8 | 52.4 | 11.3 | 43.8 | 57.1 | 7.6 | 88.0 | 82.1 | 2.8 | 35.4 | 28.2 | 1.5 | 62.5 | 54.9 | 5.8 |
| BERTScore | 79.1 | 42.9 | 20.4 | 51.5 | 35.7 | 15.2 | 88.6 | 82.1 | 11.6 | 48.7 | 33.3 | 5.3 | 67.0 | 48.5 | 13.1 |
| PRISM | **98.9** | 81.0 | 27.8 | 77.8 | 64.3 | 23.3 | 80.7 | 56.4 | 12.7 | 49.0 | 30.8 | 10.0 | 76.6 | 58.1 | 18.5 |
| BARTScore | 91.9 | **90.5** | 25.0 | 54.0 | 50.0 | 20.8 | 86.7 | 79.5 | 17.9 | 43.2 | 30.8 | 9.5 | 69.0 | 62.7 | 18.3 |
| T5SCORE-B$_{un}$ | 94.6 | 71.4 | 23.4 | 52.2 | 42.9 | 20.2 | 82.8 | 64.1 | 13.3 | 49.0 | 35.9 | 10.4 | 69.7 | 53.6 | 16.8 |
| T5SCORE-L$_{un}$ | 95.1 | 81.0 | 25.6 | 57.5 | 42.9 | 21.8 | 84.2 | 64.1 | 15.1 | 51.4 | 35.9 | 10.4 | 72.0 | 56.0 | 18.2 |
| T5SCORE-XL$_{un}$ | 93.4 | 81.0 | 27.7 | 66.6 | 64.3 | 22.7 | 86.4 | 71.8 | 16.2 | 51.7 | 41.0 | 10.5 | 74.5 | 64.5 | 19.3 |
| SUPERVISED METHODS | | | | | | | | | | | | | | | |
| BLEURT | 95.5 | 71.4 | 30.7 | **91.6** | **78.6** | 24.3 | 79.7 | 64.1 | 16.4 | 47.3 | 41.0 | 10.3 | 78.5 | 63.8 | 20.4 |
| COMET | 96.5 | 71.4 | 27.7 | 88.9 | 71.4 | 22.1 | 77.1 | 66.7 | 19.8 | 55.9 | 33.3 | **14.1** | 79.6 | 60.7 | 20.9 |
| T5SCORE-B$_{sup}$ | 98.4 | 71.4 | 25.5 | 65.2 | 42.9 | 21.8 | **91.7** | **79.5** | 18.0 | 56.5 | 46.2 | 11.5 | 78.0 | 60.0 | 19.2 |
| T5SCORE-L$_{sup}$ | 98.5 | 81.0 | 29.6 | 85.5 | 64.3 | 25.1 | 88.9 | 74.4 | **20.7** | 58.9 | 48.7 | 13.6 | 83.0 | 67.1 | **22.3** |
| T5SCORE-XL$_{sup}$ | **98.9** | 81.0 | **31.8** | 89.0 | **78.6** | 25.5 | 86.0 | 69.2 | 19.2 | **62.9** | **53.8** | 12.5 | **84.2** | **70.7** | **22.3** |

sults on three datasets: DA20, MQM21 and QE20.

## 6.1 Training details

Hyperparameters are kept the same as in Sec. 5.1. **DA20** Generative training is performed on MT-prism, while discriminative training is performed on the WMT DA corpus from 2017 to 2019. **MQM21** We take the models from DA20 and further train them for 2 additional epochs on MQM20. **QE20** DA20 models are further trained on the QE20 train split. The best checkpoint is picked based on its performance on QE20 development split and the results on the test split are reported.

## 6.2 Results

Table 3: Segment Kendall's Tau, system Pearson and system Kendall's Tau on the MQM21 dataset for source-based methods. The highest correlation for each language pair under each correlation method is bold.

| | en-de | | | zh-en | | |
| | sys-p | sys-k | seg-k | sys-p | sys-k | seg-k |
| COMET$_{src}^{mqm}$ | 68.5 | 46.2 | 16.2 | 50.5 | 41.0 | **10.0** |
| T5SCORE-B$_{src}^{mqm}$ | 42.9 | 23.1 | 13.0 | 53.0 | 41.0 | 6.2 |
| T5SCORE-L$_{src}^{mqm}$ | 67.6 | 48.7 | 16.9 | **60.8** | **51.3** | 8.6 |
| T5SCORE-XL$_{src}^{mqm}$ | **77.8** | **56.4** | **17.9** | 58.7 | 48.7 | 9.3 |

For MQM21, We compare our supervised model with the COMET$_{src}^{mqm}$ baseline (Rei et al., 2021).[11]

[11]COMET$_{src}^{mqm}$ is a source-based metric trained on WMT

Tab.3 illustrates both the segment level and system level results. For DA20, we compare our supervised model with COMET$_{src}$ baseline (Rei et al., 2021).[12] Results can be found at Appendix A.6. For QE20, we choose PRISM$_{qe}$ (Thompson and Post, 2020a) which is a reference free version of PRISM as the unsupervised baseline, and TransQuest (Ranasinghe et al., 2020b,a) which is the winner of WMT2020 Shared Task on Quality Estimation (Specia et al., 2021) as the supervised baseline.[13] Tab. 4 illustrates the segment Pearson correlation[14] of different evaluation metrics.

We have the following observations:

1) For MQM21, T5SCORE surpasses the baseline at both segment and system level on average. For DA20, T5SCORE is better than the baseline at the segment level on average, and better than the baseline for most language pairs at the system level. For QE20, both supervised and unsupervised T5SCORE surpass the corresponding baselines.

2) *Overall, source-based models perform worse than the reference-based models, but their differences are much smaller on MQM21 than DA20.* We

DA scores from 2017 to 2020 and adapted to MQM by fine-tuning on MQM20. We use the model wmt21-comet-qe-mqm.

[12]COMET$_{src}$ is trained on WMT DA scores from 2017 to 2019. We use the model wmt20-comet-qe-da.

[13]We use the Implementation and the model monotransquest-da-multilingual.

[14]The correlation metric follows Specia et al. (2021).

conjecture that the reason for this may be related to the human evaluation process where WMT-DA uses a mixture of source-based and reference-based annotations, while MQM uses the source.

Table 4: Segment level Pearson correlation on QE20 corpus for source-based methods. The highest correlation by unsupervised method is underlined, and the highest correlation overall is bold.

|  | si-en | ne-en | et-en | ro-en | en-de | en-zh | Avg |
|---|---|---|---|---|---|---|---|
| UNSUPERVISED METHODS | | | | | | | |
| PRISM$_{qe}$ | 2.9 | -13.6 | 69.4 | 82.9 | 46.4 | 30.4 | 36.4 |
| T5SCORE-B$_u$ | 53.0 | 55.7 | 58.1 | 77.1 | 13.5 | 21.3 | 46.4 |
| T5SCORE-L$_u$ | 59.3 | 60.4 | 64.1 | 80.0 | 20.4 | 25.9 | 51.7 |
| T5SCORE-XL$_u$ | 58.8 | 61.7 | 64.0 | 80.7 | 26.7 | 28.5 | 53.4 |
| SUPERVISED METHODS | | | | | | | |
| TQ | 58.9 | 75.5 | 76.6 | 88.0 | 42.3 | 44.1 | 64.2 |
| T5SCORE-B$_s$ | 58.1 | 69.9 | 71.8 | 83.4 | 44.7 | 41.4 | 61.6 |
| T5SCORE-L$_s$ | 58.9 | 75.5 | 76.8 | 86.4 | 49.8 | 44.0 | 65.2 |
| T5SCORE-XL$_s$ | 60.2 | 74.7 | 77.8 | 87.1 | 49.8 | 45.6 | 65.9 |

# 7 Analysis

We design analyses to better understand the mechanism of T5SCORE and its strenghs over other metrics, specifically asking three questions: Q1: *Is generative training necessary before discriminative training?* Q2: *What are the strengths and weaknesses of each evaluation metric?* Q3: *When will source-based evaluation outperforms reference-based evaluation?*

**Effectiveness of Generative Training**  In our experiments, discriminative training is based on the model trained generatively. To answer Q1, we compare the performance of discriminatively trained models with and without generative training on DA20. The results are shown in Fig. 4.[15] We observe that (1) Both T5SCORE-B and T5SCORE-L are enhanced by generative training before discriminative training under three correlation metrics (except that T5SCORE-B using segment level Kendall's Tau correlation has a little performance drop), which means that generative training is necessary to improve model performance. (2) Larger model T5SCORE-L benefits more from generative training, compared to T5SCORE-B, indicating that larger models are better at keeping knowledge from generative training.

**Multi-Dimension Analysis**  For Q2, we compare diverse evaluation metrics under different error cat-

---

[15]Appendix A.7 has detailed results of every language pair.

Figure 4: Segment Kendall's Tau, system Pearson and system Kendall's Tau correlations of different models with and without generative training on DA20. T5SCORE-B$_w$ and T5SCORE-L$_w$ are models with generative training, while T5SCORE-B$_{w/o}$ and T5SCORE-L$_{w/o}$ are models without generative training.

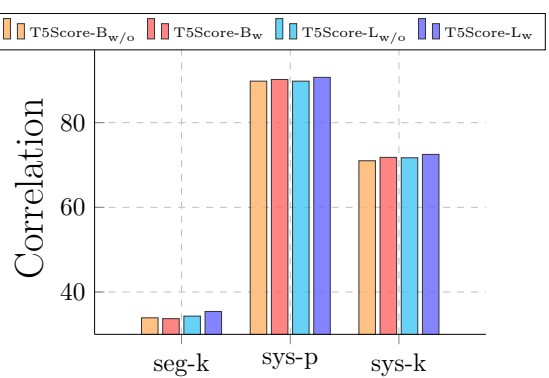

egories on the MQM datasets. To evaluate the performance of the metrics on a given error category, we use the score of each example in that error category as the gold standard to compare with the score given by the automated evaluation metrics. There are six error categories in total, including the five error categories described in Sec. 4.1 and an overall category that measures all errors. Fig. 5 shows the Root Mean Square Error (RMSE)[16] of diverse evaluation metrics under different error categories. We observe that: (1) Our model T5SCORE-XL$_{sup}$ ranks first overall in every error category except accuracy where BLEURT ranks first. (2) Supervised metrics (BLEURT, COMET, T5SCORE-XL$_{sup}$) perform better than other unsupervised metrics. (3) The evaluation perspectives that all metrics excel at are the same. All metrics perform best in terms of accuracy, much better than other error categories.

**Top-k Analysis**  To answer Q3, we conduct experiments on MQM21 and evaluate on the subset of the data from the top-k performing MT systems.[17] Results are presented in Fig. 6. We find that: (1) The advantage of the source-based version of T5SCORE over the reference-based version increases as we evaluate fewer systems, i.e., only high-quality systems. Although not as pronounced in COMET, which also uses the source for its reference-based

---

[16]We use RMSE instead of seg-k, because a certain error category only has a few examples, so the number of examples to calculate seg-k for that category will be too small to be accurate. Scores are normalized before calculating RMSE.

[17]To get more high quality translations, we also evaluate human translations in this experiment. All other experiments in this paper don't include human translations.

Figure 5: RMSE of different metrics in each error category on MQM dataset.

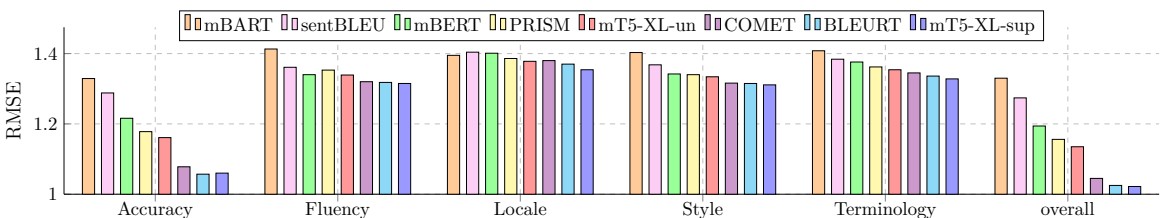

Figure 6: Metrics performance of the top-k MT systems. X-axis is number k. Y-axis is correlation results.

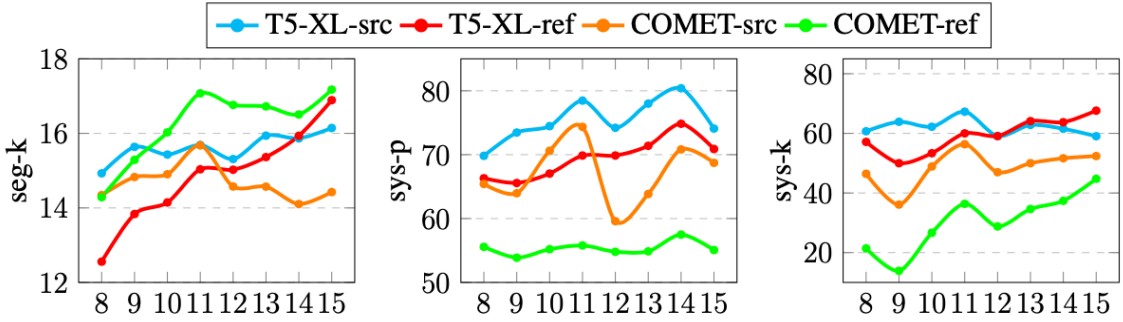

approach, it has roughly the same trend. This suggests that the source-based approach is more suitable for evaluating top systems, and that source-based evaluation should be considered as machine systems are improved. (2) T5SCORE outperforms COMET on all top-k systems under all three correlation measures, except for the reference-based version under seg-k correlation. The better performance of COMET's reference-based version may be attributed to its combination of reference and source, further indicating the importance of source.

## 8 Related Work

The increasing performance of generation systems equipped with large pre-trained models puts forward a higher requirement of the evaluation ability of automated metrics. As such, researchers are exploring different evaluation frameworks by teaching metric to learn diverse types of knowledge.

The directest way is to supervise metrics with manually annotated judgments with a trainable model, typical works include BEER (Stanojević and Sima'an, 2014), BLEURT (Sellam et al., 2020), ROBLEURT (Wan et al., 2022), COMET (Rei et al., 2020) and C-SPEC (Takahashi et al., 2021).

Despite the superior performance of optimizing the correlation with human judgments, this method is expensive in creating human annotations. To bypass this challenge, researchers attempt to evaluate generated texts in an unsupervised way by calculating the lexical or semantic similarity between reference and generated texts with surface-based string match (e.g., BLEU (Papineni et al., 2002), ROUGE (Lin, 2004) and CHRF (Popović, 2015)) or unsupervised pre-trained components (e.g., BERTSCORE (Zhang et al., 2019), YISI (Lo, 2019; Lo and Larkin, 2020) and MOVERSCORE (Zhao et al., 2019)).

Recently, works such as PRISM (Thompson and Post, 2020a) and BARTScore (Yuan et al., 2021) start to formulate evaluation as a generation task, which can not only make full use of pre-trained knowledge but also find more training data that can provide useful supervision for metrics to learn.

In this paper, we propose a framework that allows different types of signals to be incorporated into metrics. Concurrent with our work, UniEval (Zhong et al., 2022) formulates evaluation as a boolean question answering task and is trained on semi-supervised data. By contrast, our method is based on the generation formulation, which enables us to utilize a large amount of raw parallel data.

## 9 Conclusions

In this paper, we augment evaluation metrics with the ability to use different types of signal from data, which based on the assumption that a good evaluation metric not only should be informed of how to score different qualities of texts but also how high-quality texts are generated. We achieve this goal by proposing a discriminative generation frame-

work for the evaluation of generated texts, which outperform 8 existing top-performing metrics on 5 datasets from 19 languages.

## Limitations

This study has potential limitations: (1) While different perspectives (e.g. informativeness, fluency, or factuality) of text could be evaluated, we assign one overall quality score, which cannot necessarily reflect the quality of a certain aspect. In the future work, more interpretable metrics could be designed specifically for a certain evaluation perspective. (2) Due to the contrastive training objective of T5SCORE, our metric may not be as good at predicting the absolute score of segments or systems compared to predicting their relative ranks. (3) We focus on evaluating automatically generated texts by machine systems. Human generated texts which might have different features from machine generated texts could be addressed in future work. (4) We study reference-based and source-based metrics separately. A combination of both could be studied in the future. (5) The potential risk associated with our model is that the quality annotations we use may have been generated according to the preferences of a certain demographic group, for which explicit metadata is not available. As our model is trained on these annotations, it may inherit the underlying bias.

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

# A Appendix

## A.1 Training with Different Corpora

We compare our model discriminatively trained on different training corpora: T5SCORE-*[20], T5SCORE-*[21] and T5SCORE-*[21]$_{mqm}$ with baselines: COMET[20] which is trained on WMT DA corpus from 2017 to 2019 and COMET[21]$_{mqm}$ which is trained on WMT DA corpus from 2017 to 2020 and further trained for 1 additional epoch on MQM-20.[18] Tab.5 shows that adding one more year's DARR corpus (WMT-DA-20) as training data, T5SCORE-*[21] has better performance than T5SCORE-*[21]. Besides, adding MQM in training has large performance improvement, although the total number of MQM training samples are much less than samples in DA corpus.

Table 5: Segment Kendall's Tau, system Pearson and system Kendall's Tau of different metrics on MQM21 with different training data.The highest correlation for each language pair under each correlation method is bold.

| | en-de | | | zh-en | | |
|---|---|---|---|---|---|---|
| | sys-p | sys-k | seg-k | sys-p | sys-k | seg-k |
| COMET[20] | 82.3 | 64.1 | 18.4 | 37.2 | 41.0 | 11.9 |
| COMET[21]$_{mqm}$ | 77.1 | 66.7 | 19.8 | 55.9 | 33.3 | **14.1** |
| T5SCORE-B[20] | 75.8 | 53.8 | 14.5 | 42.5 | 28.2 | 10.0 |
| T5SCORE-B[21] | 82.3 | 56.4 | 16.3 | 41.6 | 28.2 | 10.9 |
| T5SCORE-B[21]$_{mqm}$ | **91.7** | **79.5** | 18.0 | 56.5 | 46.2 | 11.5 |
| T5SCORE-L[20] | 79.4 | 56.4 | 15.8 | 44.1 | 38.5 | 11.3 |
| T5SCORE-L[21] | 83.9 | 59.0 | 17.7 | 44.0 | 33.3 | 11.1 |
| T5SCORE-L[21]$_{mqm}$ | 88.9 | 74.4 | **20.7** | 58.9 | 48.7 | 13.6 |
| T5SCORE-XL[21] | 83.5 | 61.5 | 18.0 | 47.3 | 35.9 | 12.4 |
| T5SCORE-XL[20] | 78.7 | 53.8 | 16.6 | 53.4 | 43.6 | 12.2 |
| T5SCORE-XL[21]$_{mqm}$ | 86.0 | 69.2 | 19.2 | **62.9** | **53.8** | 12.5 |

## A.2 Training Details

We adopt the Adafactor (Shazeer and Stern, 2018) optimizer following Raffel et al. (2020). For generative training, our model is trained on ParaCotta, and WMT-19 is used as validation set, while WMT-20 is used as test set. The hyper-parameter tuned is learning rate. For discriminative training, on the base of the generative model, we further train our model on the z-score of WMT DA corpus from 2017 to 2019, using the discriminative loss function in equation3. The hyper-parameters tuned are learning rate, dropout rate and $\alpha$. We choose 4 language pairs (ru-en, en-ru, en-pl, en-cs) as the validation set to tune the hyper-parameters and pick the best model checkpoint.

Tab.6 shows the hyper-parameters, training time and computing resources of T5SCORE. We find as the model size increases, we need less training steps to get the best performance on the validation set, so maximum training steps is decreased

---

[18]COMET[20] uses the model wmt20-comet-da and COMET[21]$_{mqm}$ uses the model wmt21-comet-mqm.

for larger models. We use the linear learning rate scheduler and use 10% of the maximum training steps as the wamp up step. In the table, when GPU is larger than 1, we use model parallelism.

## A.3 WMT-DA segment level results

A supplement to the Tab. 1 showing the segment-level results on language pairs with English as source for the WMT DA20 corpus.

## A.4 WMT-DA system level results

Besides segment level correlation, we also calculate system level correlation on WMT DA20 corpus, and the results are shown in Tab.8,9,10,11.

## A.5 Unsupvised T5SCORE-XXL Results

In Tab.12, we show results on WMT DA20 corpus using generative T5SCORE-XXL.

## A.6 WMT-DA Source Based Evaluation Results

We conduct source-based evaluation experiments on WMT DA20 corpus using supervised T5SCORE. We compare our supervised source-based model with COMET$_{src}$ baseline (Rei et al., 2021), which is a source-based metric trained to predict WMT DA scores from 2017 to 2019.[19] Tab.13 illustrates the segment Kendall's Tau correlation of diverse evaluation metrics on WMT-DA. The results show that at segment level T5SCORE-B$_{src}$ is comparable to the baseline and larger models surpass the baseline. Our system level results, shown in Tab.14,15 are also comparable to or better than the baseline for most language pairs except iu-en and en-iu. In all tables, Avg-en denotes the average correlation across all x-en language pairs; Avg-x denotes the average correlation for all en-x language pairs; Avg denotes the average correlation for all language pairs.

## A.7 Effectiveness of Generative Training

To show the importance of unsupervised generative training, we compare the performance of T5SCORE-B and T5SCORE-L with and without unsupervised generatively training in Tab.16,17,18,19. Avg denotes the average correlation for all language pairs.

---

[19]COMET$_{src}$ uses the model wmt20-comet-qe-da.

## A.8 MultiSumm Segment Level Pearson Correlation Result

Tab.20/Tab.21 illustrates the segment Kendall's Tau/Pearson correlation of diverse evaluation methods for 8 language pairs on MultiSumm dataset.

## A.9 Dataset

### A.9.1 Evaluation Dataset

**DA20** DA20 is the Direct Assessment (DA) corpus from WMT20 metrics shared task (Mathur et al., 2020b) which includes 23,293 segments across 18 language pairs. DA20 covers 18 language pairs and 211 systems:km-en(7), en-cs(12), pl-en(14), ru-en(11), iu-en(11), en-iu(11), ta-en(14), en-pl(14), en-ta(15), zh-en(16), en-zh(12), cs-en(12), de-en(12), ja-en(10), en-ja(11), en-de(14), ps-en(6), en-ru(9).

**MQM20 & MQM21** MQM20 (Freitag et al., 2021a) and MQM21 (Freitag et al., 2021b) are datasets obtained by professional translators who re-annotated the outputs from WMT20 and WMT21 shared task according to the Multidimensional Quality Metrics (MQM) framework (Lommel et al., 2014). The MQM framework contains assessments of five aspects of the text, which are accuracy, fluency, terminology, style, and local. MQM20 covers 2 language pairs and 20 systems: en-de(10), zh-en(10) and comprises 1,418 and 2,000 segments for language pair en-de, zh-en respectively. In our experiments, we excluded 3 human translation systems for en-de and 2 human translation systems for zh-en. MQM21 covers 2 language pairs and 32 systems: en-de(17), zh-en(15) including 527 and 650 samples for en-de, zh-en respectively.

**QE20** QE20 (Specia et al., 2021) is the dataset of WMT20 shared task on Quality Estimation (QE). It covers 6 language pairs: en-de, en-zh, ro-en, et-en, ne-en, si-en, comprising 7,000/1,000/1,000 segments for Train/Dev/Test set for each language pair respectively, and each language pair uses one state-of-the-art machine translation models built using the fairseq toolkit (https://github.com/facebookresearch/fairseq).

**MultiSumm** MultiSumm (Koto et al., 2021) is a multilingual summarization dataset containing texts and their summaries in eight languages (en, id, fr, tr, zh, ru, de, es). The dataset collects 135 documents in each language, as well as summaries

Table 6: Hyper-parameters, training time and computing resources. Max-step means the maximum training steps. Save-step means we save a checkpoint every save-step. GPU shows the GPU type and the number of GPUs. A6000 is the NVIDIA RTX$^{\text{TM}}$ A6000. Time is the wall clock training time.

| | max-step | save-step | batch-size | warm-up | learning-rate | dropout | $\alpha$ | GPU | time |
|---|---|---|---|---|---|---|---|---|---|
| T5SCORE-B$_{un}$ | 10,000 | 500 | 10 | 1,000 | 5.00e-5 | 0.10 | - | A6000*1 | 0.6h |
| T5SCORE-L$_{un}$ | 10,000 | 500 | 10 | 1,000 | 5.00e-5 | 0.10 | - | A6000*1 | 1h |
| T5SCORE-XL$_{un}$ | 10,000 | 500 | 10 | 1,000 | 5.00e-5 | 0.10 | - | A6000*1 | 3h |
| T5SCORE-B$_{sup}$ | 300,000 | 10,000 | 32 | 0 | 1.00e-5 | 0.05 | 1 | A6000*1 | 60h |
| T5SCORE-L$_{sup}$ | 200,000 | 10,000 | 32 | 0 | 1.00e-5 | 0.05 | 1 | A6000*2 | 80h |
| T5SCORE-XL$_{sup}$ | 50,000 | 5,000 | 32 | 0 | 1.00e-5 | 0.05 | 1 | A6000*4 | 40h |

Table 7: Segment-level Kendall's Tau correlations on language pairs with English as source for the WMT DA20 corpus. Avg denotes the average correlation achieved by a metric across all en-x language pairs.

| | en-cs | en-de | en-iu | en-ja | en-pl | en-ru | en-ta | en-zh | Avg |
|---|---|---|---|---|---|---|---|---|---|
| UNSUPERVISED METHODS | | | | | | | | | |
| sentBLEU | 43.2 | 30.2 | 19.1 | 47.9 | 15.3 | 5.1 | 39.5 | 39.7 | 30.0 |
| BERTScore | 51.1 | 39.5 | 19.5 | 53.8 | 28.5 | 20.5 | 60.4 | 41.1 | 39.3 |
| PRISM | 61.2† | 43.7† | 19.6 | 57.8 | 39.9† | 26.8† | 39.3 | 47.3† | 41.9 |
| BARTScore | 55.9 | 42.0 | 26.0 | 57.9 | 32.1 | 24.4 | 62.8 | 46.2 | 43.4 |
| T5SCORE-B$_{un}$ | 52.4 | 39.4 | 35.8‡ | 55.9 | 31.9 | 22.4 | 62.1 | 42.8 | 42.8 |
| T5SCORE-L$_{un}$ | 56.3 | 41.1 | **35.9**‡ | 57.4 | 35.3 | 25.3 | 65.9 | 45.4 | 45.3 |
| T5SCORE-XL$_{un}$ | 58.9 | 42.2 | 34.0 | 59.8† | 38.3 | 27.4† | 66.3† | 47.1† | 46.7 |
| SUPERVISED METHODS | | | | | | | | | |
| bleurt | **69.3**‡ | 45.9 | 32.6 | 60.8 | 45.5 | 30.1 | 65.3 | 50.3 | 50.0 |
| comet | 66.8 | **46.8**‡ | 32.2 | 62.4 | **46.2**‡ | **34.5**‡ | 67.1 | **52.3**‡ | 51.0 |
| T5SCORE-B$_{sup}$ | 61.2 | 43.1 | 32.4 | 60.5 | 38.9 | 28.9 | 64.1 | 48.5 | 47.2 |
| T5SCORE-L$_{sup}$ | 66.6 | 45.6 | 33.4 | 62.8‡ | 43.9 | 31.8 | 66.7 | 52.2‡ | 50.4 |
| T5SCORE-XL$_{sup}$ | 68.0 | **46.8**‡ | 34.1 | **63.0**‡ | 45.6 | 33.8‡ | **68.1**‡ | 51.7‡ | **51.4** |

Table 8: System level Pearson correlations on language pairs with English as target for the WMT DA20 corpus. Avg. denotes the average correlation across all x-en language pairs.

| | cs-en | de-en | iu-en | ja-en | km-en | pl-en | ps-en | ru-en | ta-en | zh-en | Avg |
|---|---|---|---|---|---|---|---|---|---|---|---|
| UNSUPERVISED METHODS | | | | | | | | | | | |
| sentBLEU | 84.4 | 97.8 | 65.2 | 97.4 | 96.9 | 50.1 | 88.8 | 91.7 | 92.5 | 94.8 | 85.9 |
| BERTScore | 81.6 | 99.8 | 73.0 | 97.4 | 95.0 | 59.2 | 93.0 | 92.7 | 89.5 | 96.1 | 87.7 |
| PRISM | 81.8 | 99.8 | 83.3 | 97.4 | 95.0 | 50.2 | 96.6 | 90.8 | 89.8 | 95.7 | 88.0 |
| BARTScore | 84.4 | 99.8 | 79.2 | 97.7 | 94.4 | 52.7 | 95.1 | 92.4 | 91.4 | 95.8 | 88.3 |
| T5SCORE-B$_{un}$ | 86.1 | 99.9 | 80.9 | 98.1 | 95.2 | 52.9 | 95.8 | 93.1 | 90.7 | 96.1 | 88.9 |
| T5SCORE-L$_{un}$ | 84.4 | 99.9 | 83.8 | 98.1 | 95.2 | 53.6 | 96.7 | 92.9 | 89.9 | 96.2 | 89.1 |
| T5SCORE-XL$_{un}$ | 83.9 | 99.9 | 84.0 | 97.8 | 95.0 | 53.7 | 96.7 | 92.5 | 89.4 | 96.2 | 88.9 |
| SUPERVISED METHODS | | | | | | | | | | | |
| bleurt | 78.9 | 99.7 | 84.0 | 96.5 | 99.4 | 56.6 | 95.7 | 89.8 | 92.0 | 94.7 | 88.8 |
| comet | 78.3 | 99.8 | 85.2 | 96.4 | 97.1 | 59.1 | 94.1 | 92.3 | 88.0 | 95.2 | 88.6 |
| T5SCORE-B$_{sup}$ | 81.9 | 99.4 | 78.9 | 98.2 | 97.8 | 57.4 | 95.1 | 91.9 | 88.1 | 96.4 | 88.5 |
| T5SCORE-L$_{sup}$ | 80.4 | 99.3 | 77.8 | 98.0 | 98.6 | 59.4 | 95.4 | 92.2 | 89.6 | 96.6 | 88.7 |
| T5SCORE-XL$_{sup}$ | 79.3 | 99.3 | 81.7 | 97.4 | 98.3 | 58.0 | 95.9 | 92.0 | 88.4 | 95.5 | 88.6 |

Table 9: System level Pearson correlations on language pairs with English as source for the WMT DA20 corpus. Avg. denotes the average correlation across all en-x language pairs and Avg-all denotes the average correlation across all language pairs.

| | en-cs | en-de | en-iu | en-ja | en-pl | en-ru | en-ta | en-zh | Avg | Avg-all |
|---|---|---|---|---|---|---|---|---|---|---|
| **UNSUPERVISED METHODS** | | | | | | | | | | |
| sentBLEU | 84.1 | 93.4 | 13.5 | 94.6 | 95.0 | 98.1 | 88.2 | 92.8 | 82.5 | 84.4 |
| BERTScore | 88.4 | 95.5 | 68.4 | 97.0 | 91.4 | 97.2 | 95.4 | 92.4 | 90.7 | 89.1 |
| PRISM | 94.9 | 95.8 | 85.9 | 93.2 | 95.8 | 72.4 | 91.6 | 96.8 | 90.8 | 89.3 |
| BARTScore | 90.8 | 94.5 | 86.2 | 94.7 | 95.5 | 61.8 | 95.8 | 94.7 | 89.3 | 88.7 |
| T5SCORE-B$_{un}$ | 89.1 | 95.9 | 65.8 | 95.5 | 96.0 | 63.8 | 96.1 | 96.4 | 87.3 | 88.2 |
| T5SCORE-L$_{un}$ | 91.0 | 95.9 | 67.0 | 95.9 | 96.1 | 63.0 | 96.6 | 96.6 | 87.8 | 88.5 |
| T5SCORE-XL$_{un}$ | 92.4 | 95.2 | 63.2 | 94.9 | 96.4 | 70.6 | 96.7 | 97.0 | 88.3 | 88.6 |
| **SUPERVISED METHODS** | | | | | | | | | | |
| bleurt | 98.8 | 95.4 | 74.8 | 95.8 | 98.3 | 82.6 | 92.6 | 98.3 | 92.1 | 90.2 |
| comet | 97.8 | 97.2 | 86.0 | 97.4 | 98.1 | 92.5 | 94.4 | 98.2 | 95.2 | 91.5 |
| T5SCORE-B$_{sup}$ | 93.7 | 97.2 | 67.9 | 98.2 | 97.8 | 92.9 | 93.7 | 97.6 | 92.4 | 90.2 |
| T5SCORE-L$_{sup}$ | 96.4 | 97.3 | 67.4 | 97.4 | 98.0 | 94.8 | 95.9 | 97.8 | 93.1 | 90.7 |
| T5SCORE-XL$_{sup}$ | 96.8 | 96.8 | 61.3 | 95.9 | 98.3 | 95.7 | 95.0 | 98.6 | 92.3 | 90.2 |

Table 10: System level Kendall's Tau correlations on language pairs with English as target for the WMT DA20 corpus. Avg. denotes the average correlation across all x-en language pairs.

| | cs-en | de-en | iu-en | ja-en | km-en | pl-en | ps-en | ru-en | ta-en | zh-en | Avg |
|---|---|---|---|---|---|---|---|---|---|---|---|
| **UNSUPERVISED METHODS** | | | | | | | | | | | |
| sentBLEU | 78.8 | 75.8 | 45.5 | 73.3 | 61.9 | 27.5 | 60.0 | 60.0 | 69.2 | 85.0 | 63.7 |
| BERTScore | 78.8 | 72.7 | 63.6 | 73.3 | 71.4 | 47.3 | 86.7 | 52.7 | 67.0 | 80.0 | 69.4 |
| PRISM | 75.8 | 72.7 | 67.3 | 86.7 | 71.4 | 34.1 | 86.7 | 56.4 | 64.8 | 80.0 | 69.6 |
| BARTScore | 81.8 | 75.8 | 67.3 | 86.7 | 71.4 | 36.3 | 86.7 | 56.4 | 64.8 | 83.3 | 71.0 |
| T5SCORE-B$_{un}$ | 84.8 | 72.7 | 70.9 | 82.2 | 71.4 | 36.3 | 86.7 | 56.4 | 67.0 | 81.7 | 71.0 |
| T5SCORE-L$_{un}$ | 81.8 | 75.8 | 74.5 | 86.7 | 71.4 | 36.3 | 86.7 | 56.4 | 64.8 | 80.0 | 71.4 |
| T5SCORE-XL$_{un}$ | 78.8 | 75.8 | 74.5 | 82.2 | 71.4 | 36.3 | 86.7 | 56.4 | 62.6 | 80.0 | 70.5 |
| **SUPERVISED METHODS** | | | | | | | | | | | |
| bleurt | 75.8 | 81.8 | 63.6 | 77.8 | 100.0 | 42.9 | 86.7 | 49.1 | 60.4 | 73.3 | 71.1 |
| comet | 72.7 | 75.8 | 63.6 | 77.8 | 100.0 | 40.7 | 86.7 | 56.4 | 62.6 | 73.3 | 71.0 |
| T5SCORE-B$_{sup}$ | 72.7 | 75.8 | 56.4 | 82.2 | 90.5 | 42.9 | 86.7 | 56.4 | 60.4 | 83.3 | 70.7 |
| T5SCORE-L$_{sup}$ | 75.8 | 78.8 | 56.4 | 77.8 | 100.0 | 42.9 | 86.7 | 52.7 | 60.4 | 71.7 | 70.3 |
| T5SCORE-XL$_{sup}$ | 78.8 | 78.8 | 67.3 | 82.2 | 90.5 | 45.1 | 86.7 | 52.7 | 62.6 | 76.7 | 72.1 |

Table 11: System level Kendall's Tau correlations on language pairs with English as source for the WMT DA20 corpus. Avg. denotes the average correlation across all en-x language pairs and Avg-all denotes the average correlation across all language pairs.

| | en-cs | en-de | en-iu | en-ja | en-pl | en-ru | en-ta | en-zh | Avg | Avg-all |
|---|---|---|---|---|---|---|---|---|---|---|
| **UNSUPERVISED METHODS** | | | | | | | | | | |
| sentBLEU | 51.5 | 80.2 | 23.6 | 85.5 | 60.4 | 94.4 | 86.7 | 72.7 | 69.4 | 66.2 |
| BERTScore | 51.5 | 80.2 | 34.5 | 85.5 | 56.0 | 94.4 | 84.8 | 72.7 | 70.0 | 69.6 |
| PRISM | 81.8 | 86.8 | 45.5 | 81.8 | 67.0 | 61.1 | 73.3 | 81.8 | 72.4 | 70.8 |
| BARTScore | 54.5 | 82.4 | 41.8 | 78.2 | 64.8 | 61.1 | 84.8 | 75.8 | 67.9 | 69.7 |
| T5SCORE-B$_{un}$ | 54.5 | 84.6 | 34.5 | 85.5 | 67.0 | 61.1 | 86.7 | 69.7 | 68.0 | 69.7 |
| T5SCORE-L$_{un}$ | 51.5 | 82.4 | 34.5 | 89.1 | 67.0 | 61.1 | 84.8 | 75.8 | 68.3 | 70.0 |
| T5SCORE-XL$_{un}$ | 51.5 | 82.4 | 27.3 | 85.5 | 67.0 | 66.7 | 84.8 | 72.7 | 67.2 | 69.0 |
| **SUPERVISED METHODS** | | | | | | | | | | |
| bleurt | 90.9 | 84.6 | 34.5 | 81.8 | 73.6 | 66.7 | 73.3 | 84.8 | 73.8 | 72.3 |
| comet | 90.9 | 84.6 | 38.2 | 74.5 | 73.6 | 72.2 | 77.1 | 81.8 | 74.1 | 72.4 |
| T5SCORE-B$_{sup}$ | 75.8 | 86.8 | 34.5 | 85.5 | 69.2 | 72.2 | 77.1 | 84.8 | 73.3 | 71.8 |
| T5SCORE-L$_{sup}$ | 90.9 | 89.0 | 30.9 | 85.5 | 69.2 | 72.2 | 79.0 | 84.8 | 75.2 | 72.5 |
| T5SCORE-XL$_{sup}$ | 90.9 | 89.0 | 23.6 | 89.1 | 71.4 | 72.2 | 77.1 | 87.9 | 75.2 | 73.5 |

Table 12: Segment Kendall's Tau, system Pearson and system Kendall's Tau correlations on all language pairs for the WMT DA20 corpus using generative T5SCORE-XXL. Avg-en denotes the average correlation across all x-en language pairs, Avg-x denotes the average correlation across all en-x language pairs, and Avg denotes the average correlation across all language pairs.

| | cs-en | de-en | iu-en | ja-en | km-en | pl-en | ps-en | ru-en | ta-en | zh-en | Avg-en |
|---|---|---|---|---|---|---|---|---|---|---|---|
| seg-k | 13.5 | 49.0 | 26.9 | 27.8 | 29.9 | 8.7 | 16.7 | 13.9 | 24.4 | 15.8 | 22.7 |
| sys-p | 83.0 | 99.9 | 83.7 | 97.6 | 95.2 | 55.2 | 96.9 | 92.4 | 90.0 | 96.2 | 89.0 |
| sys-k | 72.7 | 75.8 | 74.5 | 77.8 | 71.4 | 36.3 | 86.7 | 56.4 | 64.8 | 80.0 | 69.6 |

| | en-cs | en-de | en-iu | en-ja | en-pl | en-ru | en-ta | en-zh | Avg-x | Avg |
|---|---|---|---|---|---|---|---|---|---|---|
| seg-k | 60.4 | 43.8 | 33.8 | 60.6 | 40.0 | 26.9 | 65.3 | 47.2 | 47.3 | 33.6 |
| sys-p | 93.1 | 95.6 | 64.2 | 96.1 | 96.4 | 68.5 | 96.7 | 97.2 | 88.5 | 88.8 |
| sys-k | 60.6 | 82.4 | 27.3 | 89.1 | 64.8 | 66.7 | 82.9 | 75.8 | 68.7 | 69.2 |

Table 13: Segment level Kendall's Tau correlations on WMT DA20 corpus using source-based models. The highest correlation for each language pair is bold.

| | cs-en | de-en | iu-en | ja-en | km-en | pl-en | ps-en | ru-en | ta-en | zh-en | Avg-en |
|---|---|---|---|---|---|---|---|---|---|---|---|
| COMET$_{src}$ | **9.2** | 40.8 | **3.2** | 15.3 | 14.9 | **4.6** | 9.2 | **10.1** | 16.7 | 9.2 | 13.3 |
| T5SCORE-B$_{src}$ | 7.5 | **43.2** | 0.7 | 16.2 | 11.9 | 4.4 | 9.7 | 7.5 | 19.2 | **9.4** | 12.9 |
| T5SCORE-L$_{src}$ | 5.7 | 35.9 | 0.7 | 19.2 | 16.7 | 4.1 | 10.5 | 8.4 | 22.5 | 8.1 | 13.2 |
| T5SCORE-XL$_{src}$ | 7.1 | 41.7 | 2.6 | **22.1** | **22.5** | 3.0 | **10.6** | 7.3 | **24.7** | 7.4 | **14.9** |

| | en-cs | en-de | en-iu | en-ja | en-pl | en-ru | en-ta | en-zh | Avg-x | Avg |
|---|---|---|---|---|---|---|---|---|---|---|
| COMET$_{src}$ | 61.3 | 34.6 | -6.3 | 46.7 | 35.8 | 26.4 | 51.2 | 39.8 | 36.2 | 23.5 |
| T5SCORE-B$_{src}$ | 55.3 | 36.4 | **0.0** | 51.2 | 28.6 | 24.7 | 57.1 | 41.2 | 36.8 | 23.5 |
| T5SCORE-L$_{src}$ | 60.5 | 37.1 | -5.2 | 56.9 | 37.4 | 27.2 | 63.9 | 44.3 | 40.3 | 25.2 |
| T5SCORE-XL$_{src}$ | **62.7** | **40.0** | -2.6 | **58.7** | **37.6** | **28.2** | **66.2** | **45.6** | **42.0** | **27.0** |

Table 14: System level Pearson correlations on WMT DA20 corpus using source based models.

| | cs-en | de-en | iu-en | ja-en | km-en | pl-en | ps-en | ru-en | ta-en | zh-en | Avg-en |
|---|---|---|---|---|---|---|---|---|---|---|---|
| COMET$_{src}$ | 75.5 | 93.9 | **70.6** | 89.2 | 89.6 | 44.8 | 83.2 | 88.3 | 79.5 | 84.7 | **79.9** |
| T5SCORE-B$_{src}$ | **82.6** | **99.7** | -7.8 | 94.1 | 86.9 | 47.4 | 92.2 | 90.0 | 81.6 | **95.1** | 76.2 |
| T5SCORE-L$_{src}$ | 73.9 | **99.7** | -17.3 | **96.2** | 92.2 | **52.6** | 92.0 | 91.1 | 85.5 | 92.6 | 75.8 |
| T5SCORE-XL$_{src}$ | 73.0 | 99.5 | 1.0 | 95.9 | **99.1** | 51.3 | **94.0** | **92.0** | **87.9** | 91.5 | 78.5 |

| | en-cs | en-de | en-iu | en-ja | en-pl | en-ru | en-ta | en-zh | Avg-x | Avg |
|---|---|---|---|---|---|---|---|---|---|---|
| COMET$_{src}$ | **98.9** | 90.1 | **86.3** | 95.2 | **96.9** | 80.0 | 88.8 | **97.5** | **91.7** | **85.2** |
| T5SCORE-B$_{src}$ | 95.8 | **97.3** | 31.5 | **96.8** | 93.5 | 79.4 | 91.7 | 93.7 | 85.0 | 80.1 |
| T5SCORE-L$_{src}$ | 98.2 | 96.6 | 27.4 | **96.8** | 96.2 | 84.9 | 94.9 | 95.7 | 86.3 | 80.5 |
| T5SCORE-XL$_{src}$ | 98.5 | 96.3 | 22.4 | 96.4 | 96.8 | **88.7** | **95.2** | 96.0 | 86.3 | 82.0 |

Table 15: System level Kendall's Tau correlations on WMT DA20 corpus using source based models.

| | cs-en | de-en | iu-en | ja-en | km-en | pl-en | ps-en | ru-en | ta-en | zh-en | Avg-en |
|---|---|---|---|---|---|---|---|---|---|---|---|
| COMET$_{src}$ | 69.7 | **78.8** | **52.7** | 77.8 | **90.5** | 29.7 | 73.3 | 45.5 | 51.6 | 55.0 | **62.5** |
| T5SCORE-B$_{src}$ | 66.7 | 66.7 | -9.1 | 64.4 | 61.9 | **31.9** | **86.7** | **56.4** | 60.4 | 65.0 | 55.1 |
| T5SCORE-L$_{src}$ | **72.7** | **78.8** | -12.7 | 82.2 | 81.0 | **31.9** | 73.3 | 52.7 | 56.0 | **68.3** | 58.4 |
| T5SCORE-XL$_{src}$ | **72.7** | **78.8** | 5.5 | **86.7** | **90.5** | 29.7 | 73.3 | 52.7 | **64.8** | **68.3** | 62.3 |

| | en-cs | en-de | en-iu | en-ja | en-pl | en-ru | en-ta | en-zh | Avg-x | Avg |
|---|---|---|---|---|---|---|---|---|---|---|
| COMET$_{src}$ | 84.8 | **80.2** | **60.0** | 70.9 | **80.2** | **66.7** | 56.2 | **84.8** | **73.0** | **67.1** |
| T5SCORE-B$_{src}$ | 84.8 | 78.0 | 12.7 | **81.8** | 71.4 | 61.1 | 65.7 | 81.8 | 67.2 | 60.5 |
| T5SCORE-L$_{src}$ | **93.9** | 78.0 | 1.8 | 78.2 | 71.4 | **66.7** | 77.1 | 81.8 | 68.6 | 63.0 |
| T5SCORE-XL$_{src}$ | **93.9** | **80.2** | 1.8 | 78.2 | 67.0 | **66.7** | **82.9** | 78.8 | 68.7 | 65.1 |

Table 16: Segment Kendall's Tau, system Pearson and system Kendall's Tau correlations for WMT DA20 corpus without unsupervised training using T5SCORE-B.

| | cs-en | de-en | iu-en | ja-en | km-en | pl-en | ps-en | ru-en | ta-en | zh-en |
|---|---|---|---|---|---|---|---|---|---|---|
| seg-k | 14.0 | 48.4 | 27.8 | 28.8 | 30.7 | 10.2 | 18.0 | 13.1 | 24.4 | 16.4 |
| sys-p | 82.1 | 99.2 | 79.9 | 97.8 | 97.7 | 56.3 | 94.7 | 91.8 | 88.5 | 96.3 |
| sys-k | 72.7 | 75.8 | 56.4 | 82.2 | 90.5 | 40.7 | 86.7 | 56.4 | 62.6 | 80.0 |

| | en-cs | en-de | en-iu | en-ja | en-pl | en-ru | en-ta | en-zh | Avg |
|---|---|---|---|---|---|---|---|---|---|
| seg-k | 60.9 | 43.4 | 33.4 | 59.3 | 39.6 | 29.3 | 63.7 | 48.6 | 33.9 |
| sys-p | 93.9 | 96.7 | 64.0 | 97.2 | 97.9 | 91.5 | 93.9 | 96.9 | 89.8 |
| sys-k | 69.7 | 86.8 | 30.9 | 85.5 | 69.2 | 72.2 | 77.1 | 81.8 | 71.0 |

Table 17: Segment Kendall's Tau, system Pearson and system Kendall's Tau correlations for WMT DA20 corpus without unsupervised training using model T5SCORE-L.

| | cs-en | de-en | iu-en | ja-en | km-en | pl-en | ps-en | ru-en | ta-en | zh-en |
|---|---|---|---|---|---|---|---|---|---|---|
| seg-k | 13.2 | 48.8 | 30.5 | 28.9 | 30.7 | 9.6 | 18.4 | 14.5 | 25.6 | 16.1 |
| sys-p | 81.5 | 99.3 | 79.8 | 97.9 | 97.9 | 58.2 | 95.2 | 92.2 | 89.0 | 96.5 |
| sys-k | 72.7 | 78.8 | 56.4 | 77.8 | 90.5 | 40.7 | 86.7 | 60.0 | 60.4 | 73.3 |

| | en-cs | en-de | en-iu | en-ja | en-pl | en-ru | en-ta | en-zh | Avg |
|---|---|---|---|---|---|---|---|---|---|
| seg-k | 65.1 | 44.5 | 14.8 | 63.0 | 44.6 | 30.5 | 67.2 | 51.2 | 34.3 |
| sys-p | 96.1 | 96.9 | 54.4 | 96.8 | 98.1 | 93.1 | 95.6 | 97.2 | 89.8 |
| sys-k | 90.9 | 86.8 | 20.0 | 85.5 | 71.4 | 72.2 | 79.0 | 87.9 | 71.7 |

Table 18: Segment Kendall's Tau, system Pearson and system Kendall's Tau correlations for WMT DA20 corpus with unsupervised training using model T5SCORE-B.

|       | cs-en | de-en | iu-en | ja-en | km-en | pl-en | ps-en | ru-en | ta-en | zh-en |
|-------|-------|-------|-------|-------|-------|-------|-------|-------|-------|-------|
| seg-k | 13.9  | 48.5  | 29.2  | 28.1  | 30.3  | 9.6   | 17.4  | 13.1  | 23.9  | 15.5  |
| sys-p | 81.9  | 99.4  | 78.9  | 98.2  | 97.8  | 57.4  | 95.1  | 91.9  | 88.1  | 96.4  |
| sys-k | 72.7  | 75.8  | 56.4  | 82.2  | 90.5  | 42.9  | 86.7  | 56.4  | 60.4  | 83.3  |

|       | en-cs | en-de | en-iu | en-ja | en-pl | en-ru | en-ta | en-zh | Avg  |
|-------|-------|-------|-------|-------|-------|-------|-------|-------|------|
| seg-k | 61.2  | 43.1  | 32.4  | 60.5  | 38.9  | 28.9  | 64.1  | 48.5  | 33.7 |
| sys-p | 93.7  | 97.2  | 67.9  | 98.2  | 97.8  | 92.9  | 93.7  | 97.6  | 90.2 |
| sys-k | 75.8  | 86.8  | 34.5  | 85.5  | 69.2  | 72.2  | 77.1  | 84.8  | 71.8 |

Table 19: Segment Kendall's Tau, system Pearson and system Kendall's Tau correlations for WMT DA20 corpus with unsupervised training using T5SCORE-L.

|       | cs-en | de-en | iu-en | ja-en | km-en | pl-en | ps-en | ru-en | ta-en | zh-en |
|-------|-------|-------|-------|-------|-------|-------|-------|-------|-------|-------|
| seg-k | 14.0  | 49.3  | 28.5  | 28.9  | 30.1  | 8.3   | 17.6  | 15.3  | 25.9  | 16.3  |
| sys-p | 80.4  | 99.3  | 77.8  | 98.0  | 98.6  | 59.4  | 95.4  | 92.2  | 89.6  | 96.6  |
| sys-k | 75.8  | 78.8  | 56.4  | 77.8  | 100.0 | 42.9  | 86.7  | 52.7  | 60.4  | 71.7  |

|       | en-cs | en-de | en-iu | en-ja | en-pl | en-ru | en-ta | en-zh | Avg  |
|-------|-------|-------|-------|-------|-------|-------|-------|-------|------|
| seg-k | 66.6  | 45.6  | 33.4  | 62.8  | 43.9  | 31.8  | 66.7  | 52.2  | 35.4 |
| sys-p | 96.4  | 97.3  | 67.4  | 97.4  | 98.0  | 94.8  | 95.9  | 97.8  | 90.7 |
| sys-k | 90.9  | 89.0  | 30.9  | 85.5  | 69.2  | 72.2  | 79.0  | 84.8  | 72.5 |

Table 20: Segment level Kendall's Tau correlation on MultiSumm corpus. The highest correlation is bold

|                          | Focus |      |      |      |      |      |      |      |      | Coverage |      |      |      |      |      |      |      |      |
|--------------------------|-------|------|------|------|------|------|------|------|------|----------|------|------|------|------|------|------|------|------|
|                          | EN    | ID   | FR   | TR   | ZH   | RU   | DE   | ES   | AVG  | EN       | ID   | FR   | TR   | ZH   | RU   | DE   | ES   | AVG  |
| ROUGE1                   | **49.0** | 48.9 | 48.1 | 39.6 | 48.5 | 7.5  | 47.1 | 36.8 | 40.7 | 52.9     | 50.5 | 48.5 | 32.0 | 47.5 | 9.8  | 8.9  | 51.6 | 37.7 |
| ROUGE2                   | 34.7  | 51.1 | 36.5 | 35.4 | 50.5 | -67.7 | 35.6 | 28.4 | 25.6 | 52.9     | 41.9 | 34.0 | 30.0 | 43.4 | -66.7 | 1.3  | 20.9 | 19.7 |
| ROUGEL                   | 44.9  | 46.7 | 48.1 | 37.5 | 48.5 | 5.4  | 49.4 | 38.9 | 39.9 | 54.9     | 46.2 | 44.3 | 30.0 | 49.5 | 7.8  | 11.4 | 42.9 | 35.9 |
| COMET                    | 36.7  | 55.6 | 44.2 | 33.3 | 48.5 | 39.8 | 60.9 | 45.3 | 45.3 | 41.2     | 48.4 | 25.8 | 32.0 | 47.5 | 29.4 | 31.6 | 51.6 | 38.4 |
| BERTScore                | 34.7  | 51.1 | 48.1 | **50.0** | 36.6 | 7.5 | 63.2 | 43.2 | 41.8 | 49.0     | 52.7 | 44.3 | **38.0** | 41.4 | -29.4 | 31.6 | 49.5 | 34.6 |
| BLEURT                   | 40.8  | 51.1 | 55.8 | 43.8 | 50.5 | 57.0 | 47.1 | **49.5** | 49.4 | 45.1     | 46.2 | 42.3 | 34.0 | 45.5 | **70.6** | 21.5 | 60.4 | 45.7 |
| PRISM                    | 32.7  | 48.9 | 57.7 | 25.0 | 46.5 | **61.3** | 54.0 | 43.2 | 46.2 | 45.1     | 59.1 | 34.0 | **38.0** | 59.6 | 54.9 | 31.6 | 53.8 | 47.0 |
| BARTScore                | 32.7  | 55.6 | 63.5 | 41.7 | 54.5 | 41.9 | **67.8** | **49.5** | 50.9 | 41.2     | 44.1 | 44.3 | 36.0 | 49.5 | 52.9 | **34.2** | 60.4 | 45.3 |
| T5SCORE-B$_{un}$         | 42.9  | 51.1 | **69.2** | 43.8 | 48.5 | 52.7 | **67.8** | 43.2 | 52.4 | 51.0     | 52.7 | 44.3 | 30.0 | 47.5 | 52.9 | 26.6 | 60.4 | 45.7 |
| T5SCORE-L$_{un}$         | 26.5  | **60.0** | 67.3 | 39.6 | **56.4** | **61.3** | 63.2 | 45.3 | **52.5** | 33.3  | 59.1 | **52.6** | 28.0 | **59.6** | 64.7 | 21.5 | 60.4 | **47.4** |
| T5SCORE-XL$_{un}$        | 24.5  | 55.6 | 59.6 | 39.6 | 54.5 | 57.0 | 65.5 | **49.5** | 50.7 | 33.3     | 63.4 | 44.3 | 26.0 | 49.5 | 68.6 | 19.0 | **67.0** | 46.4 |
| T5SCORE-B$_{sup}$        | 46.9  | 44.4 | 38.5 | 35.4 | 42.6 | -3.2 | 54.0 | 30.5 | 36.1 | **58.8** | 41.9 | 32.0 | 16.0 | 39.4 | -37.3 | 16.5 | 38.5 | 25.7 |
| T5SCORE-L$_{sup}$        | 38.8  | **60.0** | 42.3 | 37.5 | 42.6 | -14.0 | 42.5 | 32.6 | 35.3 | 54.9     | 52.7 | 32.0 | 26.0 | 37.4 | -41.2 | 13.9 | 42.9 | 27.3 |
| T5SCORE-XL$_{sup}$       | 24.5  | **60.0** | 42.3 | **50.0** | 48.5 | 14.0 | 65.5 | 41.1 | 43.2 | 52.9     | **61.3** | 15.5 | 32.0 | 37.4 | -7.8 | 19.0 | 49.5 | 32.5 |

Table 21: Segment level Pearson correlation on MultiSumm corpus.

|                          | Focus |      |      |      |      |      |      |      |      | Coverage |      |      |      |      |      |      |      |      |
|--------------------------|-------|------|------|------|------|------|------|------|------|----------|------|------|------|------|------|------|------|------|
|                          | EN    | ID   | FR   | TR   | ZH   | RU   | DE   | ES   | AVG  | EN       | ID   | FR   | TR   | ZH   | RU   | DE   | ES   | AVG  |
| ROUGE1                   | 59.0  | 70.2 | 68.7 | 81.0 | 82.6 | 52.0 | 87.3 | 60.1 | 70.1 | 62.3     | 70.5 | 66.2 | 75.6 | 77.9 | 46.6 | 88.8 | 63.7 | 68.7 |
| ROUGE3                   | 53.6  | 62.8 | 68.0 | 77.8 | 78.7 | 51.6 | 85.8 | 61.2 | 68.1 | 53.5     | 65.1 | 67.3 | 73.6 | 73.6 | 47.2 | 88.2 | 64.9 | 66.9 |
| ROUGEL                   | 58.2  | 69.3 | 68.8 | 80.0 | 81.6 | 51.3 | 86.8 | 61.0 | 69.7 | 61.0     | 70.7 | 66.3 | 75.5 | 78.1 | 46.2 | 88.3 | 64.7 | 68.6 |
| COMET                    | 50.6  | 67.6 | 53.1 | 72.6 | 73.1 | 49.9 | 84.8 | 55.6 | 62.8 | 50.1     | 68.7 | 50.2 | 76.6 | 70.8 | 49.9 | 80.1 | 62.1 | 62.8 |
| BERTScore                | 58.5  | 69.8 | 71.8 | 83.2 | 77.7 | 49.4 | 89.6 | 58.0 | 69.7 | 61.7     | 71.6 | 70.2 | 80.0 | 75.9 | 39.0 | 89.4 | 66.4 | 68.9 |
| BLEURT                   | 52.4  | 61.1 | 70.5 | 79.0 | 72.6 | 55.8 | 87.4 | 66.0 | 69.1 | 59.5     | 61.7 | 71.4 | 79.0 | 75.3 | 58.7 | 87.8 | 65.2 | 71.0 |
| PRISM                    | 59.3  | 59.5 | 68.1 | 78.9 | 70.8 | 42.5 | 87.0 | 53.9 | 65.8 | 61.2     | 60.9 | 64.4 | 78.3 | 69.3 | 45.5 | 89.0 | 60.0 | 66.8 |
| BARTScore                | 61.1  | 69.0 | 72.0 | 80.3 | 75.6 | 54.6 | 89.3 | 62.5 | 70.8 | 59.9     | 67.7 | 68.9 | 80.1 | 70.9 | 57.7 | 90.7 | 66.3 | 70.7 |
| T5SCORE-B$_{un}$         | 59.0  | 64.9 | 70.8 | 79.6 | 75.6 | 50.0 | 88.7 | 58.5 | 68.9 | 59.0     | 63.9 | 68.2 | 79.4 | 71.5 | 54.9 | 90.1 | 65.7 | 69.8 |
| T5SCORE-L$_{un}$         | 58.5  | 68.6 | 71.1 | 80.6 | 76.4 | 49.3 | 89.1 | 59.1 | 69.2 | 58.2     | 68.5 | 70.1 | 81.4 | 73.0 | 55.8 | 90.4 | 66.4 | 70.8 |
| T5SCORE-XL$_{un}$        | 57.8  | 69.9 | 70.2 | 80.2 | 77.0 | 50.2 | 88.9 | 59.7 | 69.1 | 58.0     | 70.4 | 70.2 | 81.3 | 72.9 | 54.7 | 90.0 | 65.7 | 70.4 |
| T5SCORE-B$_{sup}$        | 55.7  | 63.5 | 58.9 | 70.6 | 71.9 | 26.6 | 84.4 | 42.8 | 58.7 | 51.1     | 58.9 | 49.8 | 66.3 | 69.5 | 23.0 | 84.6 | 46.3 | 55.8 |
| T5SCORE-L$_{sup}$        | 56.7  | 68.8 | 59.2 | 71.8 | 70.0 | 18.1 | 83.1 | 47.1 | 58.0 | 50.6     | 65.3 | 50.0 | 66.6 | 69.4 | 12.3 | 82.3 | 45.7 | 53.8 |
| T5SCORE-XL$_{sup}$       | 54.2  | 66.7 | 63.0 | 77.0 | 73.2 | 32.9 | 87.7 | 50.2 | 62.6 | 53.8     | 67.8 | 54.1 | 74.4 | 70.7 | 34.4 | 88.1 | 55.7 | 61.6 |

generated by 2 systems: Pointer-Generator (See et al., 2017) and BERT (Liu and Lapata, 2019; Dong et al., 2019) model.

### A.9.2 Parallel Dataset

**ParaCotta** (Aji et al., 2022) A synthetic parallel paraphrase corpus across 17 languages. We use lexical BLEU between paraphrases to filter the dataset and keep the paraphrases with low lexical BLEU, in another word, high lexically diverse.[20]

**MT-prism** (Thompson and Post, 2020b) A Machine Translation dataset includes 99.8M training sentences across 39 languages. The data sources are WikiMatrix (Schwenk et al., 2019), Global Voices,[21] EuroParl (Koehn, 2005), SETimes[22], United Nations (Eisele and Chen, 2010).

### A.10 Models of Baseline Metrics

**BLEU** We use the implementation of SacreBLEU (Post, 2018) which is licensed under the Apache 2.0 License.

**ROUGE** We use the implementation of pyrouge (https://github.com/bheinzerling/pyrouge) which is licensed under the MIT License.

**COMET** We use the estimator model wmt20-comet-da trained on WMT DA17 to DA19 unless otherwise stated in Sec.4. It is licensed under Apache-2.0 license.

**BERTScore** We use the default multilingual BERTScore model: bert-base-multilingual-cased. It is licensed under the MIT License.

**BLEURT** We use the model BLEURT-20 trained on human ratings from WMT15 to WMT19. It is licensed under the Apache-2.0 license.

**PRISM** We use the model trained on machine translation data for 39 language pairs. It is licensed under the MIT License.

**BARTScore** The original version only supports English, so we generalize it to a multilingual version BARTScore by using mBART (Liu et al., 2020) and fine-tune it on ParaCotta. It is licensed under the Apache-2.0 license.

---

[20]We keep the paraphrases with lexical BLEU in the range [0,20], while the possible range is [0,100].

[21]https://casmacat.eu/corpus/global-voices.html

[22]http://nlp.ffzg.hr/resources/corpora/setimes/