# OpenReview forum: "T5Score: Discriminative Fine-tuning of Generative Evaluation Metrics"
_EMNLP/2023/Conference — EMNLP 2023 Findings_

### Official Review · Reviewer_SSiu · 2023-08-03

**Soundness:** 3

**Excitement:**

2: Mediocre: This paper makes marginal contributions (vs non-contemporaneous work), so I would rather not see it in the conference.

**Missing References:**

Not All Errors Are Equal: Learning Text Generation Metrics using Stratified Error Synthesis. Xu et al. ACL 2021.
SESCORE2: Learning Text Generation Evaluation via Synthesizing Realistic Mistakes. Xu et al. ACL 2022.

**Paper Topic And Main Contributions:**

This paper proposes T5SCORE, a text generation evaluation metric. T5SCORE ensembles a discriminative one, which is trained on unsupervised data using contrastive learning, and a generative one, which learns a conditional distribution. The proposed metric is evaluated on machine translation task and summarization task.

**Questions For The Authors:**

1. Is there one separate model for every task and language? If so, I think the potential of mT5 is not fully utilized and the usage of T5SCORE is limited.
2. In 3.3, how to find hypothesis with higher and lower ratings? Are there many2one data pairs in the original dataset for all supervised datasets?
3. Why the author use different model in different settings? In Figure2, T5SCORE-XL is used while in figure 3, T5SCORE-L is used.
4. Why the author didn't use the newest MQM dataset, like MQM-2022?
5. The author should also evaluate some diverse generation task performance to validate the performance of the proposed metric instead of only MT and Summarization.

**Reasons To Accept:**

T5SCORE can evaluate both generation task with or without reference. Due to the fact that it's constructed on mT5, it can evaluate generated texts in different languages.

**Reasons To Reject:**

1. There is no technique novelty in this paper. For the generative metric, it just learns a distribution like traditional unsupervised task. For discriminative metric, contrastive learning is not new. For example, like SESCORE2: Learning Text Generation Evaluation via Synthesizing
Realistic Mistakes, which is also a text generation evaluation metric and used contrastive learning. The author should also cite and compare with this work.
2. The proposed method didn't compare with some new methods like SESCORE, SESCORE2.
3. The experimental results exhibit mixed performance. For example, in table2, BARTScore, COMET, BLEURT also perform best on some tasks, while T5SCORE with different settings perform best on different tasks, such as T5SCORE-B on en-de (sys-k, sys-p) and T5SCORE-L on en-de (seg-k).

**Reproducibility:**

4: Could mostly reproduce the results, but there may be some variation because of sample variance or minor variations in their interpretation of the protocol or method.

**Reviewer Confidence:**

4: Quite sure. I tried to check the important points carefully. It's unlikely, though conceivable, that I missed something that should affect my ratings.

---

> ### Author Rebuttal · Authors · 2023-08-28
>
> Thank you for your thoughtful review and valuable suggestions. We appreciate the time and effort you have invested in providing this feedback. We will address your concerns and make the necessary revisions to the paper accordingly.
>
> For the concerns:
>
> 1.We appreciate your observation regarding the lack of technique novelty in the individual components of our method. Our main contribution and novelty is indeed not in proposing the generative or discriminative training approach per se, but in presenting a comprehensive framework that synergistically combines these two approaches. This integration enables us to leverage both supervised and unsupervised signals, including parallel data and human annotations, to better understand the characteristics of high-quality generations and to differentiate between superior and inferior hypotheses. While we acknowledge that the individual components, such as contrastive learning, are not new, our work demonstrates the benefits of utilizing different kinds of signals in a unified framework, which, to the best of our knowledge, has not been studied in previous work. In particular, SESCORE2 emphasizes self-supervised training without using human-annotated ratings, while our main focus is on utilizing as many possible learning signals as possible. We will ensure to cite and compare our work with SESCORE2 in the revised version of our paper.
>
> 2.Thank you for pointing out the absence of a comparison with newer methods like SESCORE and SESCORE2, but we also kindly remind that the original version of this draft was released earlier than SESCORE2, making a direct comparison challenging at the time. Nevertheless, we appreciate the importance of benchmarking against recent advancements and will incorporate comparative analyses with these methods in our revised submission.
>
> 3.Thank you for your observation regarding the mixed performance of various metrics in our experiments. We appreciate the opportunity to clarify this aspect of our work.
> Given the comprehensive nature of our experiments—which span five datasets, 19 languages, and employ three different correlation metrics at both the segment and system levels—it is indeed challenging for a single metric to uniformly outperform others across all settings. We believe that examining the average performance across various configurations can offer a more balanced view of each metric's capabilities. For example, the average score across all language pairs in the WMT DA task (as presented in Table 1) and the average score over different years of data (as shown in Table 2) can serve as more stable indicators of system performance.
> It's important to note that our research does not claim that T5SCORE is universally superior. Rather, we carefully delineate the conditions and settings under which T5SCORE shows advantages. Thus, we concur with your observation that no single metric 'wins' in all circumstances; our paper aims to highlight under what specific conditions T5SCORE has shown to be more effective.
>
>
> For the Questions:
>
> 1.To clarify, while we do employ distinct models for each task, we utilize a single model for all languages within each task. This approach aligns with the domain-specific nature of the tasks under consideration.
>
> We believe that using different models for each task is justified for several reasons. First, different tasks often require tailored datasets to optimize performance. For instance, our model trained on MQM-21 integrates additional data sets like DA-20 and MQM-20, specifically to enhance its performance on MQM-20.
>
> Second, our comparison metrics are also in line with this approach. The baseline models we use for comparison purposes, such as COMET-20 and COMET-21-MQM, also come in multiple versions optimized for different tasks. We align our models with the respective best-performing version of each baseline model to ensure a fair and accurate comparison.
>
> Lastly, our research setting varies depending on whether we are evaluating reference-based or source-based metrics. Given these varying conditions, it would be inconsistent to use a one-size-fits-all model across all tasks and metrics.
>
> 2.We employ two strategies for identifying hypotheses with higher and lower ratings, which depend on the dataset in question.
>
> For datasets like MQM and WMT-DA, each source input is associated with multiple hypotheses that have different human-assigned ratings. This many-to-one relationship between source inputs and hypotheses makes it straightforward to identify better and worse hypotheses for a given source.
>
> For datasets like QE, each source input corresponds to exactly one hypothesis with an associated human rating, forming one-to-one pairs like (s1,h1,score1), (s2,h2,score2), etc. In this case, we randomly select two pairs and construct "better" and "worse" hypotheses based on their scores. It's worth noting that these pairs have different source inputs. Our model structure is designed to be flexible and does not strictly require that the compared hypotheses stem from the same source input.
>
> 3.Thank you for pointing out the different models used in figures 2 and 3. We developed three variations of the model: T5Score-B, T5Score-L, and T5Score-XL for our research. For each setting, we tested all three models and included the results of all of them in the appendix. However, for the main body of the paper, we opted to present the results of the best-performing model for each specific setting to make the paper more concise and easier to follow.We encourage readers interested in the performance of all models to refer to the appendix for complete results.
>
> 4.The experiments conducted for this paper were primarily carried out in 2022. At that time, the MQM-2022 dataset had not yet been released, making it unavailable for inclusion in our study.
>
> 5.We appreciate your suggestion to extend our evaluation to more diverse generation tasks. While we agree that such an extension would provide a more comprehensive validation, we were constrained by space limitations in this paper. Additionally, our work already includes a variety of experimental settings—such as multiple languages, source-based and reference-based evaluations, and both supervised and unsupervised training frameworks. Given these constraints, we chose to focus on Machine Translation and Summarization as they are representative and widely-studied tasks in the NLP community. We will certainly consider expanding the scope of tasks in future work.

---

### Official Review · Reviewer_NgGN · 2023-08-04

**Typos Grammar Style And Presentation Improvements:** None
**Soundness:** 3

**Excitement:**

4: Strong: This paper deepens the understanding of some phenomenon or lowers the barriers to an existing research direction.

**Missing References:**

I believe authors missed out on one section of work that related to the metrics evaluation in the "related works" section and given they have space; I think they should ideally include these works in the discussion and references:
(1) https://direct.mit.edu/tacl/article/doi/10.1162/tacl_a_00321/96452/BLiMP-The-Benchmark-of-Linguistic-Minimal-Pairs
(2) https://arxiv.org/abs/2210.13746
(3) https://arxiv.org/pdf/2305.11806.pdf
(4) https://aclanthology.org/2023.acl-long.795/



**Paper Topic And Main Contributions:**

While generative based metrics exist (PRISM), combining it with discriminative approach might help better differentiate between a good and a bad MT translation. The paper's work might lead to more research growth in this direction of combining the generative as well as discriminative approaches in addition to the metrics development. Also, the new proposed metric T5 score could potentially be helpful in propelling field of Machine Translation and developing better MT systems in general.



**Questions For The Authors:**

1) Can the authors please refer to previous section (a) and (b) and address these? I would be willing to update my scores if these concerns are addressed.
2) What was the checkpoint used for BLEURT? Was it BLEURT-20?

**Reasons To Accept:**

The paper is well written and is extremely structured in its approach and the experimental design. It supports its claims with ample set of experiments and results. The gains from this metric are reflected in all the 5 datasets that cover 19 languages and, in both system, and segment level settings.

Except in few instances, they explore and demonstrate gains on wide variety of tasks that include gains on MQM and DA settings. It was also interesting to see the unsupervised approach performing better than many of the existing metrics in Table 1. They do demonstrate a promising approach of combining generative and discriminative approaches that could be pursued further. Overall, I feel the paper does a good job and would love to see the paper in conference this year.


**Reasons To Reject:**

While the approach does seem to be promising, I do have some concerns and queries and it would be helpful if the authors could help resolve them:
a) The gains in the correlations that is demonstrated from generative training in Figure 4 does not seem to be huge. It would be helpful if the authors could help me understand why the gains are minor and even in case of the T5 Score-B using segment level Kendel Tau's correlation shows a negative gain which has not been addressed properly. Can the authors suggest some explanation to this?

b) I also have a question in terms of fair comparison. The model mT5-XL that outperforms the existing metric has 3.7 billion parameters and most of the other metrics have around ~500 million params. Is the comparison between existing metrics and mT5-XL fair for stating that it outperforms the existing metrics on all datasets?

Another factor that I felt to be missing but maybe out of scope of present paper is that there is not enough clarity on breakdown what accuracy, locale or style actually constitutes in MQM section. It would have been much more interesting to see how this metric performs on the different controlled linguistic perturbations. Authors could have used an existing curated dataset specifically designed from these tasks like BLIMP or DEMETR etc. to understand the granular cases where metrics work good or not.

**Reproducibility:**

5: Could easily reproduce the results.

**Reviewer Confidence:**

4: Quite sure. I tried to check the important points carefully. It's unlikely, though conceivable, that I missed something that should affect my ratings.

---

> ### Author Rebuttal · Authors · 2023-08-28
>
> Thank you for your thoughtful review and valuable suggestions. We appreciate the time and effort you have invested in providing this feedback. We will address your concerns and make the necessary revisions to the paper accordingly.
>
> Concerning a) the minor gains in correlations as illustrated in Figure 4:
> We acknowledge that Figure 4 may not clearly convey the performance improvements due to its presentation of variables (seg-k, sys-p, sys-k) with differing ranges. To provide a clearer understanding, please refer to Appendix A.7, Tables 16-19, which include detailed statistics: For T5Score-Base, the average seg-k across 18 languages changes minimally from 33.9 to 33.7 with unsupervised generative training. However, sys-p improves from 89.8 to 90.2, and sys-k from 71 to 71.8. In contrast, T5Score-L sees more robust improvements across all three metrics. The average seg-k improves from 34.3 to 35.4, sys-p from 89.8 to 90.7, and sys-k from 71.7 to 72.5.
> One possible explanation is that our generative training process is analogous to pre-training, and larger models like T5Score-L benefit more from such training. These models are better at retaining knowledge gained during generative training, whereas smaller models like T5Score-Base may not realize significant gains due to their lower capacity, potentially leading them to "forget" some knowledge.
>
>
> In response to point b):
> Your question raises a valid concern about the fairness of comparing models with different parameter counts. However, there are several aspects that, when considered, make the comparison between T5Score and the other models fair.
>
> 1. Architectural Differences: We utilize T5, an encoder-decoder model, whereas other leading models like COMET and BLEURT utilize encoder-only architectures. According to the T5 paper, an "encoder-decoder model with L + L layers has roughly the same computational cost as a language model with only L layers." Therefore, from a computational cost perspective, the comparison remains fair between an encoder-only model with an encoder-decoder model with 2x parameters despite the parameter difference.
>
> 2. Smaller Variant for Comparison: We also present results using a smaller variant, mT5-L (1.2B parameters), which is more comparable to baseline methods with ~500M parameters. mT5-L outperforms these baselines in most experimental settings, strengthening our argument for the effectiveness of our approach.
>
> For example:
> - In the reference-based MT task using the MQM dataset (Table 2), mT5-L significantly outperforms BLEURT and COMET across multiple correlation metrics.
> - For the reference-based summarization task (Figure 3), mT5-L surpasses all baselines.
> - In the source-based task on the QE20 dataset (Table 4), mT5-L also exceeds the performance of baseline methods.
>
> Regarding Question 2, we used BLEURT-20 for our experiments, which was trained on human ratings spanning from WMT15 to WMT19. This information can be found in Appendix A.10 of our paper.

---

### Official Review · Reviewer_zxyM · 2023-08-05

**Typos Grammar Style And Presentation Improvements:** outperforms --> outperform in Line 393
**Soundness:** 4

**Excitement:**

4: Strong: This paper deepens the understanding of some phenomenon or lowers the barriers to an existing research direction.

**Paper Topic And Main Contributions:**

The authors propose a novel text generation metric (T5Score) trained on the signals used by both generative and discriminative metrics. They detail the generative and discriminative training procedures for T5Score, and show that it outperforms leading metrics on segment-level outputs.

**Questions For The Authors:**

QA: In Finding (1) of 6.2 --> Top-k Analysis, you state that the advantage of the source-based version of T5Score over the reference-based version increases as you evaluate fewer systems. Looking at Figure 6, I understand "fewer systems" to mean small k --> left part of x-axis. It seems from Figure 6 that Finding (1) only holds true for the first (leftmost) subfigure in Figure 6--seg-k?

**Reasons To Accept:**

- Motivation is clear (we have discriminative and generative text generation methods; why not take advantage of both approaches?)
- Proposed solution, the T5Score metric is simple but shown to be effective.
- Authors released code and models for community use.


**Reasons To Reject:**

Confusion about Claim (1) in 6.2 -> Top-k analysis (See Question A)

**Reproducibility:**

5: Could easily reproduce the results.

**Reviewer Confidence:**

4: Quite sure. I tried to check the important points carefully. It's unlikely, though conceivable, that I missed something that should affect my ratings.

---

> ### Author Rebuttal · Authors · 2023-08-28
>
> Thank you for your thoughtful review and valuable suggestions. We appreciate the time and effort you have invested in providing this feedback. We will address your concerns and make the necessary revisions to the paper accordingly.
>
> For your question about the clarity of Section 6.2's Finding (1) regarding the Top-k Analysis. We appreciate the opportunity to clarify this point.
>
> Firstly, we acknowledge that our observation serves more as a general trend rather than a stringent claim. When comparing T5-XL-src (depicted by the blue line) with T5-XL-ref (depicted by the red line), you are correct that the trend—"source-based has more advantages over reference-based when evaluating fewer systems"—is most clearly seen in the leftmost subfigure of Figure 6, focusing on seg-k.
> While not as conspicuous, this trend does persist in the middle and rightmost subfigures as well. In the middle subfigure, the gap between T5-XL-src and T5-XL-ref is relatively larger on the left part, which evaluates 8-11 systems, and narrower on the right part, which evaluates 12-15 systems. In the rightmost subfigure, T5-XL-ref gradually surpasses T5-XL-src as more systems are evaluated. We agree that this trend is not universally consistent across all k-values (e.g. the gap of T5-XL-src and T5-XL-ref for the fewest 8 systems is not always the largest among all), but it is present to varying degrees.
>
> As for why this trend is most pronounced in the first subfigure focusing on segment-level measurement, one plausible explanation is that system-level measurements are generally more robust when considering a larger number of systems. Therefore, reducing the system number could potentially make the measurements less robust compared to segment-level metrics.
>
> Regarding COMET-src and COMET-ref, the trend is indeed less obvious. As we noted in the main text, COMET-ref employs both source and reference information, making it a different case from T5Score. Our primary claim is thus primarily based on observations from the T5Score model.
>
> We hope this clarifies any confusion, and we appreciate your thorough review which helps us improve the quality of our work.

---

### Meta-Review · Area_Chair_4KJG · 2023-09-10

**Recommendation:** 4

**Metareview:**

This paper describes a new text generation metric (T5Score) trained on the signals used by both generative and discriminative metrics. The motivation is clear. The main objective of this framework is to learn evaluation metrics based on the assumption that generative and discriminative objectives can work in concert to train a better evaluator (having the best of the two worlds).
The paper is well written.
This new framework is evaluated on 5 datasets and tasks and the new metric is compared to a various of existing metrics (BLEU, ROUGE, COMET, BERTScore, BLEURT, PRISM and BARTScore). These experiments show that this new metric outperforms leading metrics on segment-level outputs.

---

### Decision · Program_Chairs · 2023-10-07

**Decision:**

Accept-Findings

**Comment:**

This paper describes a new text generation metric (T5Score) trained on the signals used by both generative and discriminative metrics. The motivation is clear. The main objective of this framework is to learn evaluation metrics based on the assumption that generative and discriminative objectives can work in concert to train a better evaluator (having the best of the two worlds).
The paper is well written.
This new framework is evaluated on 5 datasets and tasks and the new metric is compared to a various of existing metrics (BLEU, ROUGE, COMET, BERTScore, BLEURT, PRISM and BARTScore). These experiments show that this new metric outperforms leading metrics on segment-level outputs.